# ARC-Decode: Risk-Bounded Acceptance for Sampling-Based Speculative Decoding

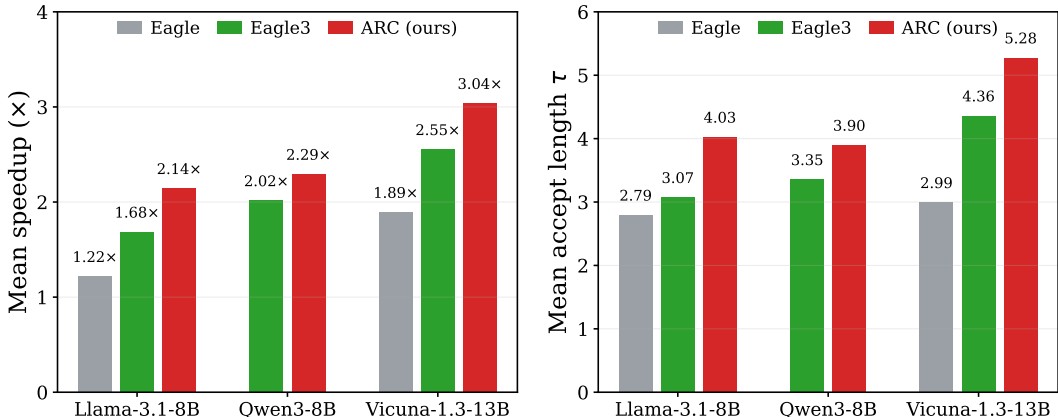

Figure 1: Average **speedup** (left) and average **accept length** $\tau$ (right) across tasks (means over MT-Bench, HumanEval, GSM8K, and Alpaca) for each model–method pair, under matched sampling settings. ARC-Decode (**ours**) consistently attains higher speedups at longer acceptance.

## ABSTRACT

As larger language models deliver stronger capabilities, their autoregressive inference becomes increasingly expensive. *Speculative decoding* accelerates generation by letting a fast draft process propose tokens that the target model verifies in parallel. Yet under sampling ($T > 0$), observed speedups consistently lag behind those under greedy decoding: verification expends compute on low-value branches, and the classical lossless verification rule rejects drafts that would induce only negligible changes in the next-step conditional distribution. A key limitation under sampling is this **over-rejection of low-risk drafts, which depresses acceptance rates and limits acceleration.** To address this gap, we propose **ARC-Decode** (**A**cceptance with **R**isk **C**ontrol), a training-free method that augments speculative decoding and requires no extra forward passes. Our method ensures relaxed acceptance while guaranteeing that accepting non–top-1 drafts causes only negligible next-step distributional shifts, as measured by Jensen–Shannon divergence. ARC-Decode combines (i) confidence-based pre-verification filtering that preserves high-probability branches while enforcing prefix closure and leaf safety, and (ii) a risk-bounded acceptance criterion using an analytic upper bound on the next-step distribution shift from embedding and logit differences. Integrated into the state-of-the-art EAGLE-3 pipeline, ARC-Decode increases accept length per cycle and reduces verification compute, achieving up to **1.6×** end-to-end speedup over EAGLE-3 under sampling with negligible quality change across benchmarks.

FIX

## 1 INTRODUCTION

Modern large language models (LLMs) demonstrate strong capabilities across tasks such as search, code generation, and dialogue (Chowdhery et al., 2023; Achiam et al., 2023). These gains follow scaling trends in model size, data, and compute, with models like Qwen3-Max (Qwen, 2025) exceeding one trillion parameters. Yet inference remains bottlenecked by autoregressive next-token

generation, enforcing sequential decoding and incurring high latency and cost (Shazeer, 2019). Reasoning-oriented workloads (Xu et al., 2025), such as GPT-o1, often produce longer and more complex contexts, increasing inference latency and motivating more efficient decoding. *Speculative decoding*(SD) addresses this by using a lightweight draft model to propose multiple tokens, which the target model verifies in parallel (Sun et al., 2023; Fu et al., 2024; Zhou et al., 2024; Li et al., 2024a). This transforms sequential generation into a partially parallel process, allowing a single forward pass to produce several outputs. By offloading draft generation and reducing memory-bound operations, SD lowers latency while maintaining the target model's generation behavior through verification. (Chen et al., 2023; Miao et al., 2024). [FIX]

While SD achieves notable speedups, we observe a significant gap between greedy and sampling modes, a discrepancy absent in standard autoregressive decoding. This gap widens as the sampling temperature increases. Medusa (Cai et al., 2024) reports that higher temperatures reduce SD efficiency due to increased rejection, even when the draft and target distributions match. Xia et al. (2024) likewise find consistent drops in acceleration as temperature rises. Across recent methods, including Speculative Sampling (Leviathan et al., 2023), EAGLE (Li et al., 2024a), HASS (Zhang et al., 2025), and EAGLE-3 (Li et al., 2025), the relative speedup under typical sampling (e.g., $T = 1$) can drop by over **20%**. This is concerning, as modern LLM applications typically rely on sampling-based generation for diversity and controllability. To understand this inefficiency, we analyze the EAGLE-3 decoding pipeline. We find that many rejected draft tokens are semantically and logically equivalent to accepted ones and yield nearly identical conditional distributions for subsequent steps (Section 3.1), suggesting that the classical lossless verification rule discards many safe draft tokens and limits speedup. These rejections reduce acceptance length and ultimately constrain the speedup [FIX] potential of speculative decoding. These observations raise a natural question:

*Can we safely increase draft token acceptance without compromising generation quality?*

To address this inefficiency, we propose ARC-Decode, a training-free and plug-in speculative decoding method that improves acceptance length under sampling regimes without additional forward passes. ARC-Decode introduces two key components: (i) an entropy-guided pre-verification pruning strategy that filters low-value draft branches using a calibrated, structure-preserving criterion (Section 3.2); and (ii) a risk-bounded relaxed acceptance rule that provably controls next-step distributional divergence (Section 3.3). Both components are designed to operate solely with verify-time information such as target logits, tied embeddings, and precomputed uncertainty scores. This ensures the method remains efficient and compatible with existing speculative sampling methods.

We apply our method to EAGLE-3 and observe consistent improvements in acceptance length and throughput across diverse models and tasks. On the Alpaca task with LLaMA3.1-8B, our method achieves up to **1.6×** speedup, and without degradation in generation quality across tasks. These results suggest that ARC-Decode can effectively enhance generation speed under sampling while reliably maintaining output quality. Our contributions are summarized as follows.

- We introduce an **entropy-guided pruning strategy** that scores draft branches using a depth-aware confidence measure combining cumulative log-probability and target entropy, effectively filtering low-value tokens while preserving valid speculative paths.
- We propose a **risk-bounded relaxed acceptance method** that certifies next-step safety via a Lipschitz-based JS bound estimated from local logit margins and pairwise embedding distances, and accepts tokens when the safety score exceeds a tunable threshold $\theta$.
- Experiments across multiple benchmarks and models show that our method consistently improves decoding speed under sampling while preserving generation quality, delivering plug-and-play acceleration in the open-source speculative decoding pipeline EAGLE-3.

## 2 RELATED WORK

**Speculative decoding** serves as an effective approach for accelerating autoregressive inference by decoupling generation into a fast draft stage and a parallel verification stage. Early variants adopt either specialized draft models (Xia et al., 2023) or scaled-down versions of the target model (speculative decoding, 2023; Leviathan et al., 2023). These typically follow a serial draft-then-verify strategy, where tokens are proposed sequentially and verified in parallel (Zhang et al., 2024). Another line of work enhances draft efficiency via tree-based decoding with improved representations.

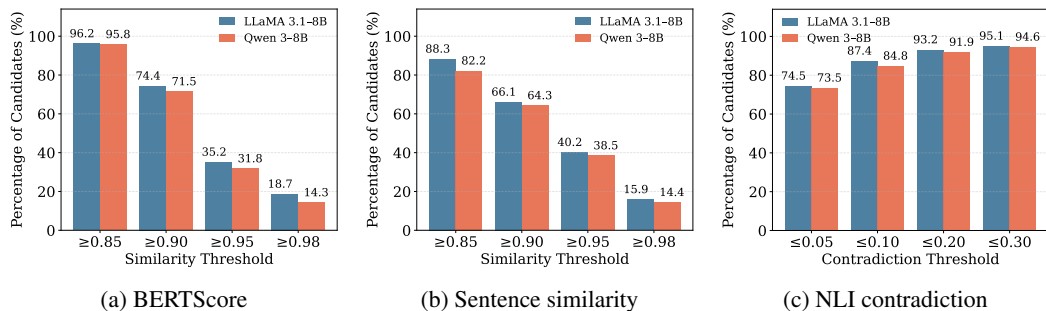

(a) BERTScore  (b) Sentence similarity  (c) NLI contradiction

Figure 2: Agreement analysis on MT-Bench at temperature $T = 1$. We compare continuations seeded by rejected draft tokens against the baseline continuation seeded by the accepted token within EAGLE-3, using two backbones: Llama-3.1-8B and Qwen-3-8B. Panels: (a) BERTScore, (b) sentence similarity, (c) NLI contradiction score. The concentration of high-agreement and low-contradiction cases indicates many rejections would not materially change subsequent generation.

Recent methods (He et al., 2024; Cai et al., 2024; Li et al., 2024a) propose multiple divergent continuations and verify them in parallel via tree attention, significantly boosting decoding throughput. To further improve draft quality, follow-up works (Zhang et al., 2025; Li et al., 2024b) use shallow draft models incorporating target model hidden states or token-level guidance for accurate and efficient multi-token prediction. **EAGLE-3** (Li et al., 2025) advances EAGLE-2 by abandoning feature-prediction constraints in favor of direct token modeling and multi-layer feature fusion, yielding higher accept length. It has been integrated into open-source frameworks such as SGLang (Zheng et al., 2024) and vLLM (Kwon et al., 2023), and is therefore adopted as our baseline.

**Limitations of Lossless Verification.** Despite advances in speculative decoding, recent work questions the necessity of strict token-level verification. **MEDUSA** (Cai et al., 2024) introduces an entropy- and probability-based acceptance mechanism that avoids exact token matching with the target model. This approach improves acceptance and efficiency while preserving quality, particularly under high-temperature sampling where traditional verification yields low acceleration due to diverse outputs. **Relaxed verification** has therefore emerged as an alternative direction, allowing safe but non-identical draft tokens to be accepted when deviations are sufficiently controlled. **Judge Decoding** (Bachmann et al., 2025) observes that even with strong draft models such as GPT-4o or LLaMA-405B, accepted spans remain short under strict verification because fluent completions that only slightly diverge from the target model are frequently rejected. This exposes a key limitation of rigid token-level matching. Judge Decoding trains a compact verifier to assess token plausibility, relaxing the acceptance criterion to allow fluent but non-identical outputs. **Fuzzy Speculative Decoding** (Holsman et al., 2025) further relaxes losslessness using a divergence threshold, though it requires computing draft–target divergence at each verified position. Together, these works highlight the limitations of strict lossless verification. ARC-Decode addresses these limitations under sampling via compute-saving pre-verification pruning and a risk-bounded next-step acceptance rule.

## 3 ARC-DECODE

This section introduces **ARC-Decode**. §3.1 analyzes speculative sampling, showing that verification dominates runtime and that many rejected drafts have negligible effect on later generation. §3.2 presents an entropy-guided pruning module, and §3.3 introduces a risk-bounded relaxed acceptance rule determining which draft tokens may be safely accepted at verification.

### 3.1 BOTTLENECKS IN SPECULATIVE DECODING UNDER SAMPLING

To identify bottlenecks, we profile the EAGLE-3 speculative decoding pipeline on MT-Bench (Llama-3.1-8B). Verification dominates runtime (**70%**; Table 1), reflecting substantial inefficiency: much compute evaluates drafts later discarded under exact matching. To assess whether these rejections are often harmless, we conduct a brief analysis via continuation experiments comparing accepted tokens with rejected alternatives. For each verification position, we force each rejected token and generate 1024-

Table 1: Runtime breakdown on MT-Bench (Llama-3.1-8B, $T{=}1$).

| Pipeline phase | Share (%) |
|---|---|
| Prefill | 3.3 |
| Draft generation | 23.5 |
| **Verification (forward pass)** | **70.4** |
| Rejection sampling | 2.8 |

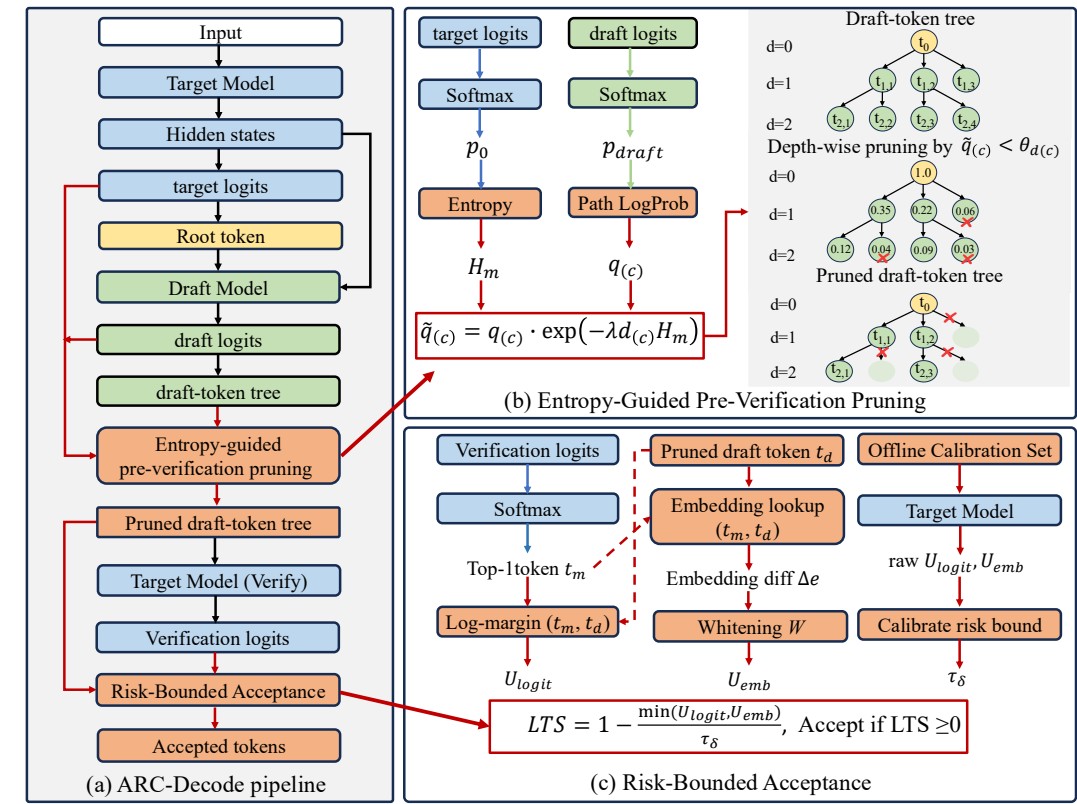

Figure 3: Diagram of the ARC-Decode inference pipeline for speculative sampling. (a) An speculative sampling pipeline with ARC-Decode steps highlighted (orange boxes). (b) *Entropy-Guided Pre-Verification Pruning*: combine entropy $H_m$ with path mass $q(c)$ to form $\tilde{q}(c)$, then prune using per-depth thresholds with prefix closure; pruned nodes are faded. (c) *Risk-Bounded Acceptance*: from verification logits and $(t_m, t_d)$, compute embedding- and logit-side bounds. Compare the tighter bound against the calibrated tolerance $\tau_\delta$ via LTS. A draft token is accepted when LTS $\geq 0$.

token continuations under identical settings. To quantify semantic agreement, we measure agreement via BERTScore (Zhang et al., 2020), MPNet-base-v2 cosine similarity (Song et al., 2020), and DeBERTa-v2 NLI contradiction (He et al., 2021), standard metrics for semantic consistency adopted in related works (Laban et al., 2022; Yang et al., 2024). The resulting distributions concentrate in high-agreement and low-contradiction regions (Fig. 2); over 70% of rejection positions satisfy this criterion, and more than half contain at least one seemingly harmless candidate.                    FIX

These findings indicate that under sampling, strict lossless verification discards many drafts whose acceptance would not affect subsequent continuations by our qualitative metrics, motivating a risk-bounded relaxed acceptance rule (§3.3). This continuation analysis is purely qualitative and not used in our acceptance criterion, which relies solely on next-step divergence bounds.

## 3.2 ENTROPY-GUIDED PRE-VERIFICATION PRUNING

To reduce the verification forward-pass overhead identified in §3.1, we prune low-value branches after drafting but before calling the target model (pipeline in Fig.3(a), pruning in Fig.3(b)).

Consider one speculation cycle with a draft tree whose nodes are indexed by $c \in \{0, \ldots, S-1\}$, with depth $d(c) \in \{0, 1, \ldots\}$ and parent pointer $\mathrm{par}(c)$. Let $\ell(c)$ be the cumulative log-probability of the draft prefix ending at node $c$, we define an entropy-aware confidence score as

$$\tilde{q}(c) = \exp\left(\ell(c)\right) \cdot \exp\left(-\lambda\, d(c)\, H_m\right), \quad \lambda \geq 0, \tag{1}$$

where the multiplicative factor reflects that acceptance risk grows with predictive uncertainty and distance from the cycle root. Using a single uncertainty scalar $H_m$ has two advantages. It is obtained

from the target prefill at no extra cost and uniformly rescales all nodes at the same depth, preserving within-depth rankings while tightening thresholds under uncertainty. Concretely, we take $H_m$ as the normalized entropy of the target model's one-step distribution at the cycle root. Let $\mathcal{V}$ denote the vocabulary with size $V = |\mathcal{V}|$, and $p_0$ be the target one-step distribution at the root:

$$H_m \;=\; \frac{-\sum_{v \in \mathcal{V}} p_0(v) \, \log p_0(v)}{\log V} \in (0, 1]. \tag{2}$$

Higher uncertainty or larger depth reduces $\tilde{q}(c)$, making the pruning more conservative. To choose thresholds, we use *per-depth mass coverage* under a risk budget $\delta \in (0, 1)$: allocate nonnegative $\{\varepsilon_d\}$ with $\sum_d \varepsilon_d \leq \delta$, and for each depth $d$ select the smallest $\theta_d$ so that the kept set $\mathcal{K}_d = \{c : d(c) = d, \; \tilde{q}(c) \geq \theta_d\}$ covers a $(1 - \varepsilon_d)$ fraction of the depth-$d$ mass:

$$\sum_{c \in \mathcal{K}_d} \tilde{q}(c) \;\geq\; (1 - \varepsilon_d) \sum_{c:\, d(c)=d} \tilde{q}(c). \tag{3}$$

We then form the kept set $\mathcal{K}$ under two structural constraints, ensuring global consistency of the retained draft tree. The first enforces *prefix closure*: every kept node retains all its ancestors, preventing inconsistencies from independent ranking and ensuring each surviving position lies on a valid root-to-leaf path within $\mathcal{K}$. For *leaf safety*, we use per-depth leaf thresholds $\tau_d$ no smaller than the corresponding depth thresholds, i.e., $\tau_d \geq \theta_d$ for all depths $d$. The constraints are defined as

$$\begin{aligned} c \in \mathcal{K} &\iff \big(\tilde{q}(c) \geq \theta_{d(c)}\big) \wedge \big(d(c) = 0 \vee \mathrm{par}(c) \in \mathcal{K}\big), \\ \mathrm{leaf}(c) \wedge \tilde{q}(c) &< \tau_{d(c)} \implies c \notin \mathcal{K}. \end{aligned} \tag{4}$$

If a kept parent would otherwise lose all children, we reinsert its highest-$\tilde{q}$ child to preserve forward extensibility and keep at least one valid continuation path. These constraints prevent stranded prefixes and maintain a contiguous high-mass backbone. This training-free pruning removes low-probability branches, reduces verification compute, and provides a structure-preserving input to the next stage. The procedure remains distinct from heuristics that adjust draft length during drafting.

## 3.3 RISK-BOUNDED ACCEPTANCE

Empirically (see §3.1), many sampling-mode rejections produce continuations that stay close to the baseline. This suggests room for relaxed acceptance if we can certify that accepting a candidate draft token will not induce a non-negligible next-step shift. We therefore develop a $\delta$-risk-bounded acceptance criterion (Fig. 3(c)) whose bound is calibrated once per backbone, depends only on verify-time quantities, and requires no additional forward passes. Intuitively, the Local Tolerance Score (LTS) uses embedding distance and logit margin to upper bound the induced next-step divergence, accepting only candidates whose predicted shift fits within a tolerable risk budget. FIX

**Problem statement.** At verification time, consider position $j$ under prefix $C$. Let $t_d^{(j)}$ be the drafted token and $t_m^{(j)}$ the target model's top-1 token under $p(\cdot \mid C)$. Here $t_m^{(j)}$ acts only as a reference for measuring the effect of substituting $t_d^{(j)}$. We assess the effect of substituting $t_d^{(j)}$ for $t_m^{(j)}$ via the target model's next-step conditional distributions:

$$q_{j+1} \;=\; p(\cdot \mid C, t_d^{(j)}), \qquad r_{j+1} \;=\; p(\cdot \mid C, t_m^{(j)}). \tag{5}$$

Our objective is to upper bound the Jensen–Shannon divergence $\mathrm{JS}(q_{j+1}, r_{j+1})$ at a chosen risk level $\delta \in (0, 1)$, thereby enabling a principled acceptance test with probabilistic control.

**Bounding next-step shift via embedding difference.** With weight tying, let $e_t \in \mathbb{R}^d$ be the input embedding of token $t$, and define the embedding difference:

$$\Delta e^{(j)} \;=\; e_{t_d^{(j)}} - e_{t_m^{(j)}}. \tag{6}$$

Let $\Phi_C : \mathbb{R}^d \to \mathbb{R}^V$ map the embedding at step $j$ to the target model's next-step logits under prefix $C$, where $V$ is the vocabulary size. Following robustness analyses of transformers (Fazlyab et al., 2019; Kim et al., 2021), we require smoothness only along the short path between the two token

embeddings, because speculative decoding changes the model input by exactly one token at this position. We therefore assume a local Lipschitz property along the segment between $e_{t_m^{(j)}}$ and $e_{t_d^{(j)}}$:

$$\left\| \Phi_C(e_{t_d^{(j)}}) - \Phi_C(e_{t_m^{(j)}}) \right\|_2 \;\leq\; \overline{L}_j \, \|\Delta e^{(j)}\|_2, \tag{7}$$

where $\overline{L}_j$ is a segment-wise average spectral bound estimated from held-out calibration traces (Appendix 1). Let $V_K$ be the size of the active vocabulary obtained from the union of top-$K$ sets, and let $L_{\mathrm{sm}} \in (0, 1]$ denote the $\ell_2$-Lipschitz constant of softmax (Kong et al., 2020). Applying softmax to the logits difference and converting the resulting $\ell_2$ bound to total variation (TV) gives:

$$\mathrm{TV}(q_{j+1}, r_{j+1}) \;\leq\; \tfrac{1}{2} \, \sqrt{V_K} \, L_{\mathrm{sm}} \, \overline{L}_j \, \|\Delta e^{(j)}\|_2. \tag{8}$$

By adding a small uniform smoothing $\mu > 0$ on the active support and applying the quadratic total variation–to–Jensen–Shannon inequality from Pang et al. (2022), we obtain:

$$\mathrm{JS}(q_{j+1}, r_{j+1}) \;\leq\; c_{\mathrm{tv}}(\mu) \, \mathrm{TV}(q_{j+1}, r_{j+1})^2 \;\leq\; c_s \, \|\Delta e^{(j)}\|_2^2, \tag{9}$$

where $c_{\mathrm{tv}}(\mu) = \mathcal{O}(1/\mu)$ and $c_s = c_{\mathrm{tv}}(\mu) \left( \frac{\sqrt{V_K} L_{\mathrm{sm}} \overline{L}_j}{2} \right)^2$. To stabilize the bound, we whiten the embedding coordinates using $W = \mathrm{diag}(1/\hat{\sigma}_1, \ldots, 1/\hat{\sigma}_d)$ with $\hat{\sigma}_k$ estimated from calibration embeddings. We absorb the constants into a factor $c_s'$ and define the embedding-side bound as

$$U_{\mathrm{emb}}^{(j)} \;=\; c_s' \, \|W \Delta e^{(j)}\|_2^2. \tag{10}$$

**Logit margin bound.** From the target logits at position $j$, let $\tilde{p}$ be post-processed probabilities after the standard logits processor and an $\epsilon$-clamp. Define the log-probability margin as

$$\Delta \tilde{\ell}^{(j)} \;=\; \log \tilde{p}(t_m^{(j)}) \;-\; \log \tilde{p}(t_d^{(j)}). \tag{11}$$

On a calibration set, we fit a scale paramete $\kappa > 0$ (e.g., via quantile regression) such that

$$\Pr\left[ \mathrm{JS}(q_{j+1}, r_{j+1}) \;\leq\; \kappa \left( \Delta \tilde{\ell}^{(j)} \right)^2 \right] \;\geq\; 1 - \delta. \tag{12}$$

We then define a logit-side upper bound using a fixed safety factor $\alpha \geq 1$, so that

$$U_{\mathrm{logit}}^{(j)} \;=\; \alpha \, \kappa \left( \Delta \tilde{\ell}^{(j)} \right)^2. \tag{13}$$

**LTS Risk-Bound Criterion.** For each position $j$, we select the tighter of the two bounds as

$$U^{(j)}(C, t_d^{(j)}) \;=\; \min\left\{ U_{\mathrm{emb}}^{(j)}, \, U_{\mathrm{logit}}^{(j)} \right\}, \qquad \Pr\left[ \mathrm{JS}(q_{j+1}, r_{j+1}) \leq U^{(j)}(C, t_d^{(j)}) \right] \;\geq\; 1 - \delta. \tag{14}$$

Let $\tau_\delta$ be the $(1 - \delta)$ quantile of $\{U^{(j)}\}$ estimated on the calibration set and fixed thereafter. Define the *Local Tolerance Score* (LTS) as

$$\mathbf{LTS}^{(j)}(C, t_d^{(j)}) \;=\; 1 \;-\; \frac{U^{(j)}(C, t_d^{(j)})}{\tau_\delta}, \tag{15}$$

which measures a safety margin relative to the risk budget (larger values indicate safer candidates). During speculative sampling, ARC-Decode only decides whether a drafted token is safe to accept; all other aspects of the sampling procedure remain unchanged. A token is accepted whenever $\mathbf{LTS}^{(j)}(C, t_d^{(j)}) \geq \theta$ (default $\theta = 0$); otherwise the step simply follows the baseline. By Theorem 1,   FIX all accepted tokens satisfy $\Pr[\mathrm{JS}(q_{j+1}, r_{j+1}) \leq \tau_\delta] \geq 1 - \delta$. The criterion is training-free and plug-in, using only verify-time target logits, tied embeddings, and calibrated constants. When combined with the pre-verification filter (§3.2), it increases acceptance length while keeping next-step shifts within a calibrated tolerance, yielding higher end-to-end efficiency in the sampling regime.

## 4 EXPERIMENTS

### 4.1 EXPERIMENTAL SETUPS

**Backbones and baselines.** We base our comparisons on four target models: Llama-3.1-8B-Instruct, Qwen-3-8B, Vicuna-13B, and Llama-3.3-70B. ARC-Decode is built on top of the EAGLE-   NEW 3 codebase and decoding pipeline (Li et al., 2025). We reuse the draft model and verification schedule from EAGLE-3, but introduce two key modifications: an entropy-guided pre-verification pruning step and a risk-bounded acceptance rule that replaces exact-match verification. All other decoding settings remain unchanged. We compare ARC-Decode with the baselines EAGLE (Li et al., 2024a), HASS (Zhang et al., 2025), Fuzzy Speculative Decoding (FSD) (Holsman et al., 2025), and EAGLE-3, and report speedup relative to vanilla autoregressive decoding.

Table 2: Experimental results on *mt-bench*, *HumanEval*, *GSM8K*, and *Alpaca*. Columns report *thrpt* ↑ (throughput; tokens/s), $\tau$ ↑ (accept length), and *speedup* ↑ (end-to-end ratio vs. vanilla) for different methods. Abbrev.: L 8B = Llama-3.1-8B, Q 8B = Qwen3-8B, V 13B = Vicuna-1.3-13B, L 70B = Llama-3.3-70B. For FSD, the parameter $T$ denotes the risk threshold (not temperature).

**NEW**

| Model | Method | MT-bench | | | HumanEval | | | GSM8K | | | Alpaca | | |
|---|---|---|---|---|---|---|---|---|---|---|---|---|---|
| | | thrpt | $\tau$ | speedup | thrpt | $\tau$ | speedup | thrpt | $\tau$ | speedup | thrpt | $\tau$ | speedup |
| L 8B | Eagle | 41.6 | 3.16 | 1.15× | 53.0 | 3.88 | 1.46× | 52.1 | 2.31 | 1.17× | 39.5 | 1.83 | 1.09× |
| L 8B | HASS | 50.2 | 2.65 | 1.38× | 62.1 | 4.32 | 1.71× | 44.8 | 2.33 | 1.24× | 49.4 | 2.51 | 1.36× |
| L 8B | Eagle3 | 66.1 | 3.35 | 1.84× | 74.4 | 3.57 | 2.05× | 51.7 | 2.52 | 1.43× | 51.6 | 2.85 | 1.42× |
| L 8B | **ARC (ours)** | **87.1** | **4.49** | **2.40×** | **77.3** | **3.94** | **2.13×** | **63.7** | **3.71** | **1.76×** | **82.7** | **3.96** | **2.28×** |
| Q 8B | Eagle3 | 61.2 | 3.00 | 1.84× | 68.1 | 3.36 | 2.06× | 66.9 | 3.88 | 2.15× | 67.3 | 3.18 | 2.01× |
| Q 8B | **ARC (ours)** | **81.6** | **4.28** | **2.45×** | **76.1** | **3.42** | **2.30×** | **68.2** | **4.13** | **2.19×** | **74.6** | **3.76** | **2.23×** |
| V 13B | Eagle | 39.9 | 2.85 | 1.82× | 43.5 | 2.97 | 1.98× | 32.7 | 3.14 | 1.74× | 45.7 | 3.01 | 2.03× |
| V 13B | Eagle3 | 60.1 | 4.01 | 2.74× | 57.6 | 3.96 | 2.62× | 41.5 | 4.69 | 2.21× | 59.3 | 4.78 | 2.64× |
| V 13B | **ARC (ours)** | **73.4** | **5.28** | **3.35×** | **71.4** | **4.74** | **3.25×** | **47.4** | **5.50** | **2.52×** | **68.1** | **5.58** | **3.03×** |
| L 70B | FSD (*T*=0.4) | 6.32 | 1.52 | 1.39× | 10.42 | 3.43 | 2.29× | 7.69 | 2.48 | 1.69× | 11.15 | 3.17 | 2.45× |
| L 70B | FSD (*T*=0.6) | 8.51 | 2.54 | 1.87× | 10.74 | 3.56 | 2.36× | 9.05 | 3.21 | 1.99× | 13.15 | 3.91 | 2.89× |
| L 70B | FSD (*T*=0.8) | 8.10 | 2.37 | 1.78× | 10.74 | 3.56 | 2.36× | 9.60 | 3.66 | 2.11× | 13.74 | 4.14 | 3.02× |
| L 70B | EAGLE-3 | 15.24 | 4.04 | 3.35× | 14.01 | 4.11 | 3.08× | 12.56 | 4.37 | 2.76× | 14.74 | 3.82 | 3.24× |
| L 70B | **ARC (ours)** | **16.38** | **4.90** | **3.60×** | **14.70** | **4.13** | **3.23×** | **12.83** | **4.68** | **2.82×** | **15.52** | **4.20** | **3.41×** |

**Tasks.** Following EAGLE and Spec-Bench (Xia et al., 2024), we evaluate four common tasks under standard evaluation settings: multi-turn dialogue, code generation, mathematical reasoning, and instruction following. We use MT-Bench (Zheng et al., 2023), HumanEval (Chen et al., 2021), GSM8K (Cobbe et al., 2021), and Alpaca (Taori et al., 2023) as the corresponding benchmarks.

**Metrics.** We evaluate decoding using both **efficiency** and **accuracy** metrics. Efficiency includes: throughput (tokens/s), i.e., wall-clock decoding speed; accepted length $\tau$, the average number of tokens accepted per verification; and speedup, the ratio over the autoregressive baseline. Accuracy is measured by four representative tasks: MT-Bench (GPT-4o–scored 1–10), HumanEval (pass@1 accuracy), GSM8K (exact match score), and Alpaca (win rate vs. GPT-4-Turbo via AlpacaEval).

**Implementation and hyperparameters.** All experiments use a temperature of 1.0. For 8B–13B models, evaluation is performed on a single NVIDIA A6000 GPU, while the 70B model is evaluated on four A6000 GPUs. We expose two pruning hyperparameters: a per-depth mass-coverage level $\varepsilon = 0.05$ (retaining the top 95% path mass at each depth) and an entropy weight $\lambda = 1.0$. A global threshold $\theta = 0.3$ is applied to the LTS score across all settings. The risk budget uses a fixed $(1-\delta) = 0.95$ quantile $\tau_\delta$, estimated on a held-out calibration set from which whitening statistics and scaling constants are also computed; all of these remain fixed during testing. Speedup is hardware-dependent, and acceptance length may vary slightly due to numerical differences. For consistency, we follow the prompt and evaluation configurations of OpenCompass (Contributors, 2023). Additional results (e.g., MMLU-Pro) and implementation details are provided in Appendix A.3.

## 4.2 EFFICIENCY AND ACCURACY RESULTS

**Efficiency.** Table 2 shows that ARC-DECODE consistently improves both acceptance length ($\tau$) and end-to-end speedup across all backbones and tasks. Under matched decoding settings, our method delivers higher throughput than EAGLE-3 and maintains stable gains across sampling workloads. On *Alpaca* with Llama-3.1-8B, ARC-Decode reaches **2.28×** speedup over vanilla, about **1.6×** faster than EAGLE-3; similar patterns appear on Vicuna-13B (e.g., *HumanEval*: 3.25× vs. 2.62×).

We also evaluate on the larger Llama-3.3-70B model. ARC-Decode attains up to **3.60×** speedup and higher acceptance lengths (e.g., $\tau$=4.90 on MT-Bench), surpassing both EAGLE-3 (**3.35×**) and the relaxed-acceptance baseline FSD, which remains below **2×** across most settings. These results indicate that the method scales effectively with model size and continues to improve efficiency.

**Accuracy.** To assess the impact of ARC on generation quality, we compare ARC with EAGLE-3 on MT-Bench, HumanEval, GSM8K, and Alpaca under matched prompts, sampling hyperparameters (e.g., temperature) and stopping criteria (Table 3). Despite replacing the verification policy, ARC

Table 3: Benchmark performance of Eagle3 and ARC-Decode on *MT-Bench*, *HumanEval*, *GSM8K*, and *Alpaca*. Metrics: *MT-Bench*: GPT-4o–judged score (1–10); *HumanEval*:pass@1(%); *GSM8K*: EM (Exact Match, %); *Alpaca*: pairwise win rate (%) vs. GPT-4-Turbo (AlpacaEval).

| Model | Method | MT-bench score ↑ | HumanEval pass@1 (%) ↑ | GSM8K EM (%) ↑ | Alpaca win rate (%) ↑ |
|-------|--------|------------------|------------------------|----------------|-----------------------|
| L 8B | Eagle3 | 6.89 | **57.2** | 77.0 | 22.8 |
| L 8B | **ARC (ours)** | **7.50** | 57.1 | **77.1** | **23.9** |
| Q 8B | Eagle3 | 6.97 | **66.3** | 75.3 | 16.3 |
| Q 8B | **ARC (ours)** | **6.99** | 66.2 | 74.9 | **17.1** |
| V 13B | Eagle3 | 6.10 | 12.1 | **23.8** | 6.6 |
| V 13B | **ARC (ours)** | **6.23** | **14.6** | 23.6 | **8.0** |

Table 4: Ablation on Llama-3.1-8B across four tasks. Columns: $\tau$ ↑ (accept length), *speedup*↑ (end-to-end throughput relative to the vanilla autoregressive baseline), and $\rho$ (pruned fraction, %). "N/A" indicates settings without pruning. All results use temperature 1.0 and matched prompts.

| Method | MT-Bench | | | HumanEval | | | GSM8K | | | Alpaca | | |
|--------|----------|---------|--------|-----------|---------|--------|-------|---------|--------|--------|---------|--------|
| | $\tau$ | speedup | $\rho$ | $\tau$ | speedup | $\rho$ | $\tau$ | speedup | $\rho$ | $\tau$ | speedup | $\rho$ |
| EAGLE-3 | 2.92 | 1.71× | N/A | 3.57 | 2.05× | N/A | 2.52 | 1.32× | N/A | 2.85 | 1.42× | N/A |
| ARC (prune-only) | 2.96 | 1.74× | 24.2 | 3.62 | 2.04× | 16.1 | 2.64 | 1.35× | 15.9 | 2.86 | 1.42× | 21.4 |
| ARC (LTS-only) | 4.21 | 2.28× | N/A | 3.85 | 2.08× | N/A | 3.65 | 1.72× | N/A | 3.82 | 2.06× | N/A |

attains accuracy on par with EAGLE-3 across all reported configurations and tasks, with parity on most metrics and occasional small gains (e.g., Llama-3.1-8B on MT-Bench: 7.50 vs. 6.89; Vicuna-13B on MT-Bench: 6.23 vs. 6.10). Overall, ARC preserves generation quality while delivering efficiency gains, exhibiting stable behavior across backbones, datasets, and decoding settings.

## 4.3 ABLATION STUDIES

We conduct ablations on Llama-3.1-8B at $T = 1.0$ across all four tasks, evaluating *prune-only* (entropy-guided pre-verification pruning; no LTS) and *LTS-only* (risk-bounded acceptance; no pruning) under identical prompts, stopping criteria, and decoding settings. Results are in Table 4.

**Prune-only.** Speedup gains over EAGLE-3 are limited: since verification runs in parallel, latency remains similar. Pruning removes low-mass branches before verification, reducing compute and filtering low-quality paths, particularly effective for wide trees or high-entropy contexts.

**LTS-only.** LTS is the primary driver of speedup: it increases $\tau$ and throughput on all tasks by certifying drafted tokens that remain within the calibrated tolerance. In full ARC-Decode framework, pruning removes low-confidence branches before verification, while LTS extends accepted prefixes, reducing cycles per token and maximizing end-to-end efficiency.

## 4.4 SENSITIVITY ANALYSES

**Temperature sensitivity.** We evaluate ARC-Decode across temperatures $T \in \{0.1, 0.3, 0.5, 0.7, 0.9\}$ (Fig. 4). As $T$ increases, both accept length $\tau$ and speedup gradually decline (e.g., MT-Bench: $\tau$ : 5.72→4.78; speedup: 3.13×→2.66×). Higher temperatures flatten the target distribution, raise entropy, and enlarge the effective branching factor, making pruning more conservative and reducing draft–target alignment. We additionally include EAGLE-3 (gray curves). ARC-Decode outperforms EAGLE-3 at every temperature on all benchmarks, and the gap widens as $T$ grows. When $T$ is low, generation is nearly deterministic and both methods behave similarly. As $T$ increases, sampling variability widens the discrepancy between the draft and target distributions, making exact-match acceptance increasingly unreliable. EAGLE-3 therefore becomes brittle under high-temperature sampling, whereas ARC-Decode's risk-bounded criterion remains more robust and continues to certify plausible drafts. Overall, ARC-Decode delivers consistently higher $\tau$ and speedup and degrades far more gracefully under high-temperature sampling. [NEW]

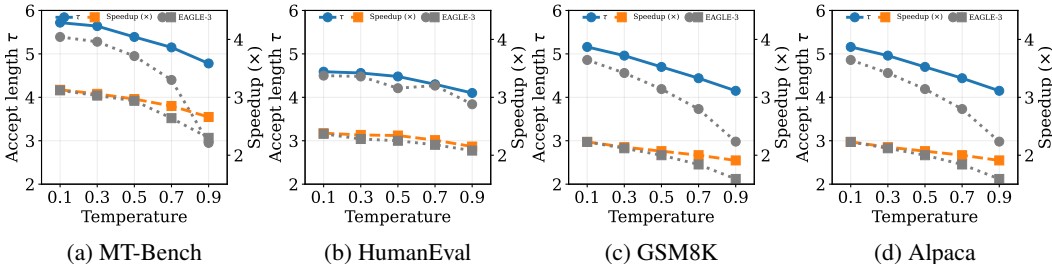

(a) MT-Bench     (b) HumanEval     (c) GSM8K     (d) Alpaca

Figure 4: Temperature sensitivity of ARC on Llama-3.1-8B across four tasks. We report accept length $\tau$ (blue) and speedup (orange), measured relative to the vanilla autoregressive baseline under temperatures $T \in \{0.1, 0.3, 0.5, 0.7, 0.9\}$. Gray curves denote the EAGLE-3 baseline.

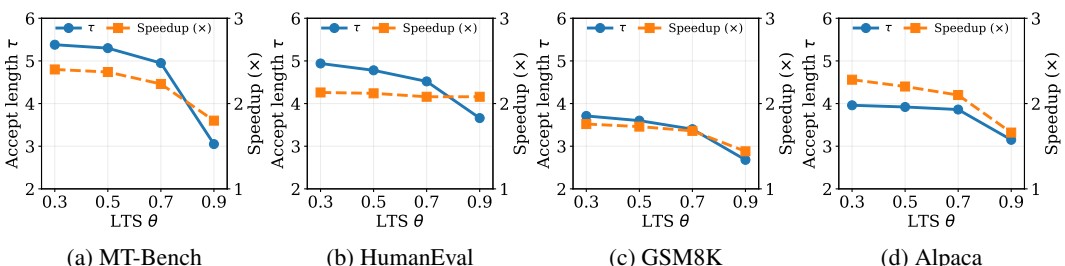

(a) MT-Bench     (b) HumanEval     (c) GSM8K     (d) Alpaca

Figure 5: Sensitivity to LTS threshold $\theta$ on Llama-3.1-8B across four tasks: accept length $\tau$ (blue) and speedup (orange), reported relative to a vanilla autoregressive baseline. All runs use matched prompts and stopping criteria; $\theta \in \{0.3, 0.5, 0.7, 0.9\}$.

**LTS-threshold sensitivity.** We evaluate the LTS threshold $\theta$ on Llama-3.1-8B, sweeping $\theta \in 0.3, 0.5, 0.7, 0.9$ and report $\tau$ and speedup in Fig. 5. As $\theta$ increases, both metrics decline, with a sharp drop between $0.7$ and $0.9$. Since LTS $= 1 - U/\tau_\delta$, a token is accepted only when $U \leq (1 - \theta)\tau_\delta$. At higher $\theta$ values (e.g., $\theta > 0.7$), this constraint becomes stringent, requiring very small embedding and logit differences for acceptance. This tight acceptance criterion disproportionately penalizes otherwise valid drafts with minor discrepancies, leading to reduced $\tau$. Because $\theta$ acts as a safety margin and no accuracy degradation was observed, we adopt $\theta = 0.3$ as a default.

ARC-Decode consistently improves speed across diverse tasks and backbones while preserving generation accuracy. As shown in Table 4, ablation studies demonstrate that LTS is the main driver of efficiency, significantly boosting acceptance length and throughput without additional computation. Pruning alone contributes moderate gains by removing low-mass paths, and further enhances speedup when combined with LTS. Sensitivity analysis (Fig. 4, 5) confirms the robustness of ARC-Decode across temperatures and threshold choices. Speedup gradually declines with increasing temperature or stricter thresholds, but remains consistently strong in practical settings. These results validate the flexibility and stability of ARC-Decode across key generation parameters. Further implementation details, calibration setup, and extended results are included in Appendix A.3.

## 5 CONCLUSION

Speculative decoding accelerates LLM inference by proposing draft tokens in parallel and verifying them with the target model. While effective under greedy decoding, its performance degrades under sampling. This work follows the emerging line of *relaxed* speculative decoding, which aims to retain more informative drafted tokens while keeping the target distribution under control. To address this, we present **ARC-Decode**, a training-free, plug-in method that enlarges accepted prefixes with calibrated next-step risk. It combines (i) entropy-guided pre-verification pruning with prefix closure, which removes low-confidence draft branches while preserving extendable paths, and (ii) a Local Tolerance Score that accepts drafts when an analytic upper bound on next-step JS divergence falls below a calibrated threshold. Integrated into EAGLE-3, ARC-Decode increases acceptance length and yields up to **1.6**× speedup over EAGLE-3 on MT-Bench, HumanEval, GSM8K, and Alpaca across multiple backbones, without measurable accuracy loss. Ablations verify the contributions of pruning and LTS, and sensitivity analysis shows robustness across temperatures. Overall, ARC-Decode provides consistent speedups under sampling without sacrificing accuracy.

## 6  REPRODUCIBILITY STATEMENT

We provide details to support reproducibility:

- **Method and Inference Setup:** Sections §3 and §4 describe all model variants, decoding configurations (e.g., temperature, top-$k$, draft depth), and evaluation protocols.
- **Hyperparameters:** All relevant hyperparameters used in decoding are reported in Section §4 and Appendix A.3.
- **Datasets:** All datasets used in our experiments are publicly available. Usage details, pre-processing steps, and licensing information are summarized in Section §4.
- **Theoretical Derivations:** Assumptions, derivations, and proofs for theoretical results are included in Section §3 and Appendix A.1.
- **Code:** We will release the codebase upon publication to facilitate reproducibility.

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

# A APPENDIX

## A.1 GUARANTEE AND PROOF

**Setup and notation.** At verification position $j$ under accepted prefix $C$, let

$$q_{j+1} = p(\cdot \mid C, t_d^{(j)}), \qquad r_{j+1} = p(\cdot \mid C, t_m^{(j)})$$

denote the target model's next–step conditionals after substituting a draft token $t_d^{(j)}$ for the model top-1 token $t_m^{(j)}$. We measure discrepancy by $\mathrm{JS}(q_{j+1}, r_{j+1})$. With weight tying, let $e_t \in \mathbb{R}^d$ be the input embedding of token $t$ and $\Delta e^{(j)} = e_{t_d^{(j)}} - e_{t_m^{(j)}}$. Let $\Phi_C : \mathbb{R}^d \to \mathbb{R}^V$ map the token embedding at step $j$ to the next–step logits. On the active vocabulary obtained by the union of top-$K$ sets we apply a small uniform smoothing $\mu > 0$ and renormalize.

**Assumptions.** (i) *Distributional Stability Under Embedding Perturbation.* We adopt a standard local Lipschitz regularity assumption: the embedding-level perturbation introduced at verification position $j$ induces a controlled and stable change in the next-step distribution. This distributional stability is sufficient for deriving the JS bound:

$$\left\| \Phi_C(e_{t_d^{(j)}}) - \Phi_C(e_{t_m^{(j)}}) \right\|_2 \le \overline{L}_j \, \|\Delta e^{(j)}\|_2, \tag{16}$$

where $\overline{L}_j$ denotes a local Lipschitz sensitivity coefficient estimated from calibration, providing a tight upper bound on how embedding perturbations propagate to the next-step logits.

(ii) *Softmax Lipschitz Continuity.* The softmax operator is $L_{\mathrm{sm}} \in (0, 1]$–Lipschitz in $\ell_2$, ensuring that a small logit perturbation leads to a proportionally bounded change in the output distribution. On the restricted support of size $V_K$, we additionally use $\|x\|_1 \le \sqrt{V_K}\|x\|_2$ to move from an $\ell_2$ bound on logits to an $\ell_1$ (TV) bound on probabilities.

(iii) *Smoothed TV–JS quadratic bound.* Following standard results on the local behavior of $f$-divergences under lower-bounded densities (Arjovsky et al., 2017), if the mixture $m = \frac{1}{2}(q + r)$ satisfies $\min_i m_i \ge \mu$, then the JS divergence admits a stable quadratic upper bound in terms of TV:

$$\mathrm{JS}(q, r) \le c_{\mathrm{tv}}(\mu) \, \mathrm{TV}(q, r)^2, \qquad c_{\mathrm{tv}}(\mu) \le \frac{1}{2\mu}. \tag{17}$$

This condition ensures that the JS discrepancy grows quadratically with small perturbations.

(iv) *Calibration–test consistency.* Since both calibration traces and test-time positions are generated by the same target model under matched decoding settings, their conditional distributions (e.g., conditioned on depth or root entropy) are aligned. Consequently, the empirical $(1 - \delta_2)$ quantile computed on the calibration set can be directly reused at test time as a valid risk-control threshold.

**Embedding-side deterministic bound.** Combining Equation (16) with assumptions yields

$$\mathrm{JS}(q_{j+1}, r_{j+1}) \le c_s \, \|\Delta e^{(j)}\|_2^2, \qquad c_s \equiv c_{\mathrm{tv}}(\mu) \left( \frac{\sqrt{V_K} L_{\mathrm{sm}}}{2} \right)^2 \overline{L}_j^2. \tag{18}$$

After applying diagonal whitening $W = \mathrm{diag}(1/\hat{\sigma}_1, \ldots, 1/\hat{\sigma}_d)$ on a held-out calibration set and absorbing constants into $c_s'$, the embedding-side surrogate used by LTS becomes

$$U_{\mathrm{emb}}^{(j)} = c_s' \, \|W \Delta e^{(j)}\|_2^2, \tag{19}$$

which provides a deterministic, position-wise upper bound on the next-step JS divergence.

**Logit-side high-probability bound.** At verification stage, let $\tilde{p}$ denote the post-processed probabilities after the standard logits processor and an $\epsilon$-clamp, which ensures numerical stability for small margins. Define the verify-time log-probability margin

$$\Delta \tilde{\ell}^{(j)} = \log \tilde{p}(t_m^{(j)}) - \log \tilde{p}(t_d^{(j)}). \tag{20}$$

Using calibration traces drawn under the decoding setup, we fit a constant $\kappa > 0$ such that

$$\Pr\left[ \mathrm{JS}(q_{j+1}, r_{j+1}) \le \kappa \, (\Delta \tilde{\ell}^{(j)})^2 \right] \ge 1 - \delta_1. \tag{21}$$

With a fixed conservative safety factor $\alpha \ge 1$, the resulting logit-side surrogate used at inference is

$$U_{\mathrm{logit}}^{(j)} = \alpha \, \kappa \, (\Delta \tilde{\ell}^{(j)})^2. \tag{22}$$

**Combined bound and empirical quantile.** To obtain a practical acceptance rule at verification time, we combine the embedding- and logit-side surrogates into a single per-position upper bound:

$$U^{(j)} \;=\; \min\{U^{(j)}_{\text{emb}},\; U^{(j)}_{\text{logit}}\}. \tag{23}$$

On the calibration traces, we compute the empirical $(1 - \delta_2)$ quantile $\tau_{\delta_2}$ of $\{U^{(j)}\}$ and fix it for all subsequent inference. By calibration–test consistency (Assumption (iv)),

$$\Pr\big[\,U^{(j)} > \tau_{\delta_2}\,\big] \;\leq\; \delta_2, \qquad j \in \mathcal{J}_{\text{test}}, \tag{24}$$

The Local Tolerance Score (LTS) gate therefore accepts at position $j$ whenever $U^{(j)} \leq \tau_{\delta_2}$.

**Theorem 1** (Risk bounds under LTS gating). *Suppose Equation* (21) *holds on the calibration set with level $\delta_1$, and the verification rule accepts whenever $\min\{U^{(j)}_{\text{emb}},\; U^{(j)}_{\text{logit}}\} \leq \tau_{\delta_2}$. Let $\tau = \tau_{\delta_2}$. Then for any verification position $j \in \mathcal{J}_{\text{test}}$ the following hold:*

(i) *Conditional risk bound (per acceptance).*

$$\Pr\big[\text{JS}(q_{j+1}, r_{j+1}) > \tau \,\big|\, U^{(j)} \leq \tau\big] \;\leq\; \delta_1.$$

(ii) *Unconditional tail bound.*

$$\Pr\big[\text{JS}(q_{j+1}, r_{j+1}) > \tau\big] \;\leq\; \delta_1 + \delta_2.$$

*Proof.* Let $A = \{U^{(j)}_{\text{emb}} \leq \tau\}$, $B = \{U^{(j)}_{\text{emb}} > \tau,\; U^{(j)}_{\text{logit}} \leq \tau\}$, and $C = \{U^{(j)} > \tau\}$. By construction the acceptance event is the disjoint union $A \cup B$.

**(i) Conditional bound.** On $A$ we have deterministically $\text{JS} \leq U^{(j)}_{\text{emb}} \leq \tau$ by equation 19, hence $\Pr(\text{JS} > \tau \mid A) = 0$. On $B$, using equation 21 and the fact that $\alpha \geq 1$,

$$\Pr\big(\text{JS} \leq U^{(j)}_{\text{logit}} \leq \tau \mid B\big) \;\geq\; 1 - \delta_1,$$

and therefore $\Pr(\text{JS} > \tau \mid B) \leq \delta_1$. Using total probability over $A \cup B$ proves the claim.

**(ii) Unconditional bound.** Bounding $\Pr(\text{JS} > \tau)$ is obtained by:

$$\Pr(\text{JS} > \tau) = \Pr(\text{JS} > \tau, A) + \Pr(\text{JS} > \tau, B) + \Pr(\text{JS} > \tau, C).$$

The contribution from $A$ is zero by determinism. The part involving $B$ is controlled by the calibration bound $\delta_1 \Pr(B) \leq \delta_1$. For the component associated with $C$, equation 24 ensures

$$\Pr(\text{JS} > \tau, C) \leq \Pr(C) = \Pr(U^{(j)} > \tau) \leq \delta_2.$$

Combining these bounds yields the unconditional guarantee $\delta_1 + \delta_2$. $\qquad\square$

**Corollaries and practical variants.** (i) *Bucketed calibration.* If calibration and test traces are only conditionally exchangeable given features (e.g., depth, root entropy, prompt length), perform the above procedure per bucket; the guarantees then hold conditionally within each bucket.

(ii) *Sequence-level control.* For a sequence of $T$ verification positions, applying a union bound to the unconditional tail bound yields $\Pr\big[\exists j \leq T : \text{JS}(q_{j+1}, r_{j+1}) > \tau_{\delta_2}\big] \leq T(\delta_1 + \delta_2)$. The overall risk budget can be allocated across positions to keep a fixed target level.

(iii) *Finite-sample correction.* The empirical quantile can be replaced by a conservative binomial (e.g., Clopper–Pearson or Wilson) bound to control $\delta_2$ in the small-calibration regime.

**Remarks on constants and complexity.** All constants $(c'_s, \kappa, \tau_{\delta_2})$ are estimated once on the calibration set and then fixed. At inference, computing $U^{(j)}_{\text{logit}}$ is $O(1)$ and computing $U^{(j)}_{\text{emb}}$ is $O(Kd)$, reusing verify-time quantities and requiring no extra forward passes.

*Calibration stability.* Because both surrogates depend only on the target model's embedding space and verify-time logits, which are intrinsic properties of the backbone, the calibrated constants transfer across different domains and prompt styles used in our experiments. *Intuitively, LTS controls divergence in the model's own next-step distribution; as long as the backbone, vocabulary, and embedding geometry remain fixed, the underlying divergence structure remains unchanged.* NEW

---

**Algorithm 1** Calibration of LTS (Local Tolerance Score) parameters

---

**Require:** target model $f$, tied embeddings $E$, logits processor $g$, calibration corpus $\mathcal{D}_{\text{cal}}$, risk level $\delta$, vocab size $K$

1: Compute per-dimension inverse std from $E$ and set $W = \text{diag}(1/\hat{\sigma})$
2: Initialize lists $\mathcal{S}_{\text{JS}}$, $\mathcal{S}_{\text{emb}}$, $\mathcal{S}_{\text{logit}}$
3: **for** each sample $(C, j)$ in $\mathcal{D}_{\text{cal}}$ **do**
4:     Compute target distribution $\tilde{p} = g(f(C))$ at position $j$
5:     Let $t_m = \arg\max \tilde{p}$ and choose a candidate $t_d \neq t_m$ (e.g., uniformly among top-$K$ excluding $t_m$)
6:     Set $U_{\text{emb,raw}} = \|W(e_{t_d} - e_{t_m})\|_2^2$,   $U_{\text{logit,raw}} = (\log \tilde{p}(t_m) - \log \tilde{p}(t_d))^2$
7:     Form next-step distributions $q_{j+1} = f(C \frown t_d)$ and $r_{j+1} = f(C \frown t_m)$ restricted to the top-$K$ union of their supports
8:     Compute JS over this union
9:     Append JS to $\mathcal{S}_{\text{JS}}$, $U_{\text{emb,raw}}$ to $\mathcal{S}_{\text{emb}}$, $U_{\text{logit,raw}}$ to $\mathcal{S}_{\text{logit}}$
10: **end for**
11: $c_s' \leftarrow \text{Quantile}_{1-\delta}(\{\text{JS}/U_{\text{emb,raw}}\})$
12: $\alpha\kappa \leftarrow \text{Quantile}_{1-\delta}(\{\text{JS}/U_{\text{logit,raw}}\})$
13: For each item, define $U_{\min}^{(j)} = \min\{c_s' U_{\text{emb,raw}}^{(j)}, \alpha\kappa\, U_{\text{logit,raw}}^{(j)}\}$
14: $\tau_\delta \leftarrow \text{Quantile}_{1-\delta}(\{U_{\min}^{(j)}\})$
15: **return** $(W, c_s', \alpha\kappa, \tau_\delta)$

---

## A.2   LTS Implementation Details

**Estimating constants used by the bound.** We treat the theoretical inequalities in Equations (7), (9) and (10) as given (see §A.1) and focus on how the required constants are calibrated once and then reused at inference. For the segment-average Lipschitz factor $\overline{L}_j$ in Equation (7), we estimate a surrogate on a held-out calibration set via finite differences along a few points $s \in [0, 1]$ on the segment $e_{t_m^{(j)}} + s\,\Delta e^{(j)}$, aggregated per bucket (e.g., by depth, root entropy, or $\|\Delta e\|_2$). A high quantile of these bucketwise values yields a conservative estimate, absorbed into $c_s$ and $c_s'$ in Equations (9) and (10). For the TV→JS coefficient $c_{\text{tv}}(\mu)$, we fix the active-vocabulary size $K$ and smoothing $\mu > 0$ and treat the resulting constant as tied to $(K, \mu)$. Since all quantities depend only on the backbone's embedding geometry and its next-step logits, the calibrated constants are inherently *per-backbone* and do not vary with downstream task domains.

**Calibration protocol and pseudocode.** For verified position $j$, we compute: (i) a near-oracle next-step divergence $\text{JS}(q_{j+1}, r_{j+1})$ on the top-$K$ union, (ii) an embedding-side raw score $U_{\text{emb,raw}}^{(j)} = \|W \Delta e^{(j)}\|_2^2$ with diagonal whitening, and (iii) a logit-side raw score $U_{\text{logit,raw}}^{(j)} = (\Delta \tilde{\ell}^{(j)})^2$ from post-processed probabilities. The construction of token pairs in calibration is solely for estimating local variation in the next-step distribution and does not prescribe how acceptance decisions are made at inference time, ensuring context-agnostic applicability.

**Deployment.** At verification time we compute

$$U_{\text{emb}}^{(j)} = c_s' \|W \Delta e^{(j)}\|_2^2, \qquad U_{\text{logit}}^{(j)} = \alpha\kappa\,(\Delta\tilde{\ell}^{(j)})^2, \qquad U^{(j)} = \min\{U_{\text{emb}}^{(j)}, U_{\text{logit}}^{(j)}\}.$$

Then $\text{LTS}^{(j)} = 1 - U^{(j)}/\tau_\delta$, and acceptance occurs when $\text{LTS}^{(j)} \geq \theta$. Since all calibrated constants depend only on backbone-level geometric and probabilistic structure, the same parameters transfer across different domains without retraining. No additional forward passes are required.

## A.3   Additional Experiments

### A.3.1   Experimental Details

**Decoding and speculative setup.** Unless noted otherwise, decoding uses temperature $T = 1.0$ with the unmodified EAGLE–3 verification schedule. We set the draft-tree depth to $S = 6$ and

Table 5: Empirical tightness of the JS upper bound across four benchmarks. Coverage is the fraction of accepted positions satisfying $\text{JS} \leq U_{\min}$; Wilson 95% confidence intervals are shown.

| Task | Coverage (JS $\leq U_{\min}$) | 95% CI (Wilson) |
|------|------------------------------|-----------------|
| MT-Bench | **99.4%** | [98.7, 99.7] |
| HumanEval | **95.7%** | [93.1, 98.4] |
| GSM8K | **95.3%** | [91.7, 98.1] |
| Alpaca | **97.6%** | [96.5, 98.4] |

NEW

cap the number of drafted tokens per cycle at `max_draft_tokens`= 32, with top-$k$ sampling following the backbone defaults. For Qwen-3–8B we disable the optional "thinking" mode.

*Choice of draft-length.* Our hardware environment differs from the high-throughput settings used in official EAGLE–3 evaluations: all experiments are run on NVIDIA RTX A6000 (48 GB), whose memory bandwidth and compute throughput make large draft trees significantly more latency-sensitive. As observed in prior analyses of speculative decoding efficiency Tang et al. (2025), increasing the draft length enlarges the tree, KV-cache footprint, and attention/masking cost, and these overheads are not always amortized on memory-bound GPUs. Under this constraint, a limit of 32 drafted tokens yields more stable end-to-end latency while keeping the comparison between EAGLE–3 and ARC-Decode fair by using a fixed tree size across all methods.

FIX

**ARC-Decode settings.** We use a global LTS threshold $\theta = 0.3$. All other ARC-Decode components follow the configurations described in the main text and Appendix, with calibration performed once and then frozen. All remaining decoding settings are kept identical to EAGLE-3.

**Calibration set.** We calibrate all ARC–Decode constants once per backbone using a 200-prompt subset of the public OpenAssistant OASST1 dataset (English, single-turn). Prompts are uniformly sampled (fixed seed) with target lengths in [32, 256], providing a diverse collection of local decoding contexts while remaining disjoint from our evaluation domains. Calibration reuses the same decoding traces produced during standard speculative verification and introduces no additional model evaluations. All other decoding settings match those of EAGLE–3.

### A.3.2 Empirical Validation of LTS Risk Coverage

To evaluate the empirical tightness of the upper bounds used by LTS, we audit accepted positions across four benchmarks (MT-Bench, HumanEval, GSM8K, Alpaca) using the Llama-3.1-8B backbone under the same decoding configuration as in §4. For each accepted draft token, we compute the true next-step JS divergence $\text{JS}(q_{t+1}, |, p_{t+1})$ and compare it against the operational bound $U_{\min} = \min U_{\text{emb}}, U_{\text{logit}}$. This analysis is performed offline from decoding traces and does not affect inference. Across all datasets, the operational bound reliably over-approximates the true next-step divergence, with coverage consistently above 95% and reaching 99.4% on MT-Bench. These results confirm that $U_{\min}$ provides a stable and accurate surrogate for local distributional sensitivity, supporting the validity of the risk-bounded acceptance rule used by ARC-Decode.

### A.3.3 Additional Comparison with Medusa and Judge Decoding

We additionally compare ARC-Decode with two representative relaxed-acceptance baselines: *Medusa* and *Judge Decoding*. For Medusa (Cai et al., 2024), we use the official pretrained `Medusa-1` and `Medusa-2` checkpoints, trained on Vicuna-13B and Vicuna-13B-v1.5, respectively. For Judge Decoding (Bachmann et al., 2025), since no pretrained model is available, we re-implement its judge classifier following the procedure described in the original paper, pairing the EAGLE-3 draft model with Llama-3.1-8B as the target model. All methods are evaluated on MT-Bench and Alpaca under identical decoding hyperparameters and prompting setups, using a single NVIDIA RTX A6000 GPU. Across both datasets and model scales, ARC-Decode yields higher accept length and higher end-to-end speedup than Medusa and Judge Decoding.

Table 6: Comparison with Medusa and Judge Decoding across two datasets.

| Model | Method | MT-Bench | | Alpaca | |
|---|---|---|---|---|---|
| | | Accept | Speedup | Accept | Speedup |
| Llama-3.1-8B | Judge | 4.17 | 2.01 | 3.15 | 1.74 |
| | EAGLE-3 | 3.35 | 1.84 | 2.85 | 1.42 |
| | **ARC (Ours)** | **4.49** | **2.40** | **3.96** | **2.28** |
| Vicuna-13B | Medusa-1 | 2.58 | 2.13 | 2.62 | 2.16 |
| | Medusa-2 | 3.26 | 2.65 | 3.24 | 2.64 |
| | EAGLE-3 | 4.01 | 2.74 | 4.78 | 2.64 |
| | **ARC (Ours)** | **5.28** | **3.35** | **5.58** | **3.03** |

NEW

Table 7: Performance on the challenging MMLU-Pro benchmark. ARC-Decode maintains accuracy while providing higher accept length and speedup.

| Method | Accept Length | Speedup | Accuracy (%) |
|---|---|---|---|
| Vanilla | — | $1.00\times$ | **32.4** |
| EAGLE-3 | 2.81 | $1.60\times$ | **32.3** |
| **ARC (Ours)** | **3.50** | $\mathbf{1.88\times}$ | **32.3** |

NEW

### A.3.4 Evaluation on a More Challenging Benchmark: MMLU-Pro

To assess the robustness of ARC-Decode under substantially more difficult reasoning workloads, we additionally evaluate all methods on **MMLU-Pro** (Wang et al., 2024), a large-scale benchmark containing 12,032 queries and task difficulty than those used in our main experiments. We compare vanilla Llama-3.1-8B, EAGLE-3, and ARC-Decode under identical decoding settings (temperature $= 1.0$, no chain-of-thought prompting, no few-shot examples).

Across this challenging dataset, ARC-Decode matches the accuracy of vanilla decoding and EAGLE-3 while achieving higher accept length and greater end-to-end speedup. These results indicate that ARC-Decode maintains output performance even on difficult, high-mismatch tasks.

### A.3.5 Comparison with Alternative Pruning Rules

To assess the effect of different draft-tree pruning rules, we compare our entropy-guided pruning (Sec.3.2) with several heuristic alternatives under the same EAGLE-3 setup (Qwen3-8B, temperature 1.0) on MT-Bench, using a single NVIDIA RTXA6000 GPU. All methods share the same verification rule of EAGLE-3, only the pruning policy differs.

- **Entropy-guided (ours).** Entropy–depth scoring with per-layer mass control, prefix-closure, and leaf-safety.
- **Depth.** Linear depth-biased ranking.
- **Depth-Exp.** Exponential depth bias favoring deeper nodes.
- **Accum-Prob.** Cumulative branch probability along the root-to-node path.

Entropy-guided pruning achieves the highest acceptance length and throughput among all tested strategies. Depth-based and cumulative-probability heuristics are less effective because they make node-wise decisions without enforcing consistent root-to-leaf paths, which limits usable tree capacity. In contrast, our rule maintains prefix-consistent paths and selects drafts more efficiently.

### A.4 Prompt Templates Used in Evaluation

NEW

We follow the standard OpenCompass (Contributors, 2023) prompting setup. Prompts are rendered using each model's chat template as invoked internally by `apply_chat_template()` in Open-Compass. Unless otherwise noted, all tasks use single-turn prompts without few-shot examples or chain-of-thought prompting (and Qwen-3-8B is evaluated without the "thinking" mode).

Table 8: Effect of alternative pruning strategies on Qwen3-8B+EAGLE-3 decoding (MT-Bench).

| Pruning strategy | Accept length ↑ | Throughput (tok/s) ↑ |
|---|---|---|
| Ours (entropy-guided) | **2.70** | **56.4** |
| Depth (best $\alpha$) | 2.53 | 51.2 |
| Depth-Exp | 2.53 | 51.1 |
| Accum-Prob | 2.57 | 51.8 |

NEW

**MT-Bench.** Rendered using the model's chat template. System prompt:

> You are a helpful, respectful and honest assistant.

**HumanEval.** System prompt:

> You are a helpful, honest assistant. You write Python **function bodies only** (no `def` line and no comments).

User prompt: the official `prompt` string from HumanEval.

**GSM8K.** We use the **4-shot** prompt format. User message:

> Question: {question}
> Let's think step by step
> Answer:

**Alpaca.** System prompt:

> You are a helpful, respectful and honest assistant.

User prompt: the instruction text from AlpacaEval.

**MMLU-Pro.** System prompt:

> You are a helpful, respectful and honest assistant, using the given options to choose the single best answer.

User prompt:

> You are given a multiple-choice question. Choose the single best answer.
> Question: {question}
> Options: A. {opt1}, B. {opt2}, C. {opt3}, ...

Answers are restricted to a single letter (A, B, C, ...).

### A.5 STATEMENT ON THE USE OF LARGE LANGUAGE MODELS (LLMS)

Large language models were used solely for grammar correction. All scientific content was developed and validated independently by the authors.

### A.6 ETHICS STATEMENT

The research adheres to the ICLR Code of Ethics. All experiments use publicly available pretrained language model weights and standard academic benchmarks, with no access to personally identifiable information or sensitive data. The work focuses on improving the computational efficiency of autoregressive inference in large language models and does not introduce new capabilities that could enable misuse. The proposed method is a training-free, plug-in optimization that does not modify base model weights and constrains distributional shifts within a calibrated risk budget, thus posing no additional risks related to fairness, safety, or societal impact.

