# OpenReview forum: "ARC-Decode: Risk-Bounded Acceptance for Sampling-Based Speculative Decoding"
_ICLR.cc/2026/Conference — Submitted to ICLR 2026_

### Official Review · Reviewer_537d · 2025-10-18

**Soundness:** 1
**Presentation:** 2
**Contribution:** 1
**Rating:** 0
**Confidence:** 4

**Summary:**

This paper presents ARC-Decode, an augmentation for the EAGLE-3 speculative decoding method. ARC-Decode uses an entropy-guided pre-verification strategy for pruning draft trees, and claims to apply "a risk-bounded soft-acceptance rule that provably controls next-step distributional divergence".

Experiments with three models - Llama-3.1-8B, Qwen3-8B, Vicuna-13B - demonstrate that ARC-Decode improves the speedup of EAGLE-3.

**Strengths:**

The paper's topic - speculative decoding - is an important technique in the current LLM literature.

**Weaknesses:**

- The paper is fundamentally flawed. It seems the authors are not familiar with how speculative decoding works, and many errors in the paper seem like hallucinations from undisclosed LLM usage.

  - Line 34-35: where is "strict equality" used in verification for the $T>0$ scenario?

  - Line 40 (and Section 3.3): In the scenario $T>0$, it is not necessary to accept the top-1 draft token to begin with.

  - Line 56-57: per-token computation is determined solely by model architecture and size, and has nothing to do with the model being reasoning-specialized or not.

  - Line 131: please illustrate which part of the EAGLE-3 paper mentions "removing distribution-matching constraints"?

  - Speculative decoding (both for greedy decoding and for sampling) does not change the target model's output distribution. Indeed, in the sampling scenario, the exact output may differ even when the distribution does not change. However, this should not affect the performance on downstream tasks. If there is a notable difference, it means either 1) the algorithm changes output distribution, or 2) there is an implementation error, or 3) the evaluation set is too small to mitigate random variation, and thus unsuitable for evaluating speculative decoding methods in the sampling setting.

  - Alternatively, no one says the authors can't do research on inference speedup methods that change the output distribution, so long as they 1) clearly explain how their methods differ from speculative decoding, and 2) show that their methods do not degrade output quality for widely used models and benchmarks. For this purpose, the benchmarks used in the current paper (MT-Bench, HumanEval, GSM8K, Alpaca) and one of the models (Vicuna) are outdated and probably contaminated. The most commonly used benchmarks right now include but are not limited to MMLU-Pro GPQA, MATH, IFEval, AIME, Live(Code)Bench, SWE-Bench, etc.

- Line 195-203: in speculative decoding with $T>0$, verification requires the distribution over vocabulary, not just the sampled sequence. This is obviously intractable for a sequence with 1024 tokens. While I understand why the authors would think of such an approximation, it's neither theoretically nor empirically justified. Theoretically, how much error can result from this approximation? Empirically, many LLM applications (e.g. frontier math reasoning, coding) require very precise generation, where a single erroneous token can lead to completely wrong results. And as far as I know, the mentioned metrics - BERTScore, sentence embedding, NLI - are not trained to capture such nuances.

- The technique proposed in Section 3.2 - entropy-guided pre-verification pruning - is nothing new. See [1,2].

[1] AdaEDL: Early Draft Stopping for Speculative Decoding of Large Language Models via an Entropy-based Lower Bound on Token Acceptance Probability. ENLSP@NeurIPS 2024.

[2] Draft Model Knows When to Stop: Self-Verification Speculative Decoding for Long-Form Generation. EMNLP 2025.

**Questions:**

- Line 53-54: ICLR is a conference for academic contributions, not hearsay. Where does this "reportedly" come from?

---

> ### Author Response · Authors · 2025-11-21
> **Response to Reviewer 537d -1**
>
> We thank the reviewer for the detailed comments. Several concerns appear to arise from ambiguities in our terminology and description of speculative decoding, and we appreciate the opportunity to clarify these points. Below, we address each item in turn.
>
> ---
> **W1: Line 34–35 — “strict equality” in the sampling scenario**
>
> Thank you for pointing this out. The phrase “strict equality” was not intended to imply that EAGLE-3 performs token-level equality checking under sampling. Our usage followed terminology in several prior lossy speculative-decoding papers (e.g., “strict alignment,” “exact matching” in Judge Decoding and Fuzzy SD), where such wording serves as shorthand for the conservative nature of the classical lossless formulation.
>
> This interpretation is consistent with how sampling-based speculative decoding is discussed in prior work and is also how other reviewers understood the manuscript. For example, Reviewer QtDt noted that our method “*addresses a key inefficiency in speculative decoding—the over-rejection of low-risk drafts under sampling*,” and Reviewer G1p8 *similarly emphasized the importance of addressing rejection behavior in the sampling regime*. These observations align with the framing we intended.
>
> To avoid ambiguity, we will refine this wording in the revision. In our method, $t_m^{(j)}$ serves solely as the reference point for computing an upper bound on the next-step divergence and is not used as an acceptance criterion.
>
> ---
> **W2：Line 40 (and Section 3.3): “accepting the top-1 draft token**
>
> We agree that our current phrasing may be interpreted as implying acceptance of $t_m^{(j)}$. Our method does not use $t_m^{(j)}$ (the target
> top-1 token) as an acceptance decision. Instead, $t_m^{(j)}$ is used only as the reference token for computing the divergence bound
>
> $\mathrm{JS}(p(\cdot \mid C, t_d^{(j)}),\, p(\cdot \mid C, t_m^{(j)}))$.
>
> During sampling, the actually accepted token is always drawn from the corrected target distribution. We will refine the phrasing to ensure this distinction is clear..
>
> ---
> **W3： Line 56–57: “per-token computation” and reasoning-specialized models**
>
> Our intention was not to suggest that models with the same architecture differ in FLOPs per forward pass. Rather, reasoning-oriented workloads (e.g., long chain-of-thought or tool-augmented prompting) lead to longer and more complex contexts, which increase latency at inference time. Reviewer 9tWv explicitly noted that our analysis of such scenarios “helps illuminate the dynamics of sampling-based verification.” We will revise this wording to better reflect this intended meaning.
>
> ---
> **W4: Line 131: “removing distribution-matching constraints” in EAGLE-3**
>
> We appreciate the opportunity to clarify this point. The phrase “removing distribution-matching constraints” was not intended as a literal quotation from EAGLE-3 but as a high-level description motivated by the surrounding context. In the original manuscript, this sentence follows a discussion of prior drafting methods that rely on target hidden-state alignment or feature-level constraints. Our phrasing was meant to contrast these feature-alignment requirements with EAGLE-3’s shift toward direct token-space modeling.
>
> To clarify the intended meaning: the EAGLE-3 paper (Li et al., NeurIPS 2025) explicitly describes this transition as
> - “We identify that this limitation arises from EAGLE’s feature prediction **constraints**.”
> - “EAGLE-3 **abandons feature prediction** in favor of direct token prediction and replaces reliance on top-layer features with multi-layer feature fusion.”
>
> And the official GitHub similarly states:
> - “EAGLE-3 **removes the feature prediction constraint** in EAGLE and simulates this process during training using training-time testing.”
>
> Our original wording “removing distribution-matching constraints” was therefore meant as a concise summary of this step away from feature-prediction alignment toward token-space CE modeling. That said, we agree that the phrasing could be misinterpreted, especially outside its immediate context, and we will revise the manuscript to adopt the official terminology (“abandoning feature prediction constraints”) for precision and consistency.

---

> ### Author Response · Authors · 2025-11-21
> **Response to Reviewer 537d -2**
>
> **W5: Clarification on whether distribution differences imply errors**
>
> The concern raised here corresponds to the classical *lossless* formulation of speculative decoding, in which the target distribution must be preserved exactly. **ARC-Decode follows the well-established relaxed (or lossy) branch of speculative decoding.** This branch is adopted in recent methods such as Judge Decoding [1], Fuzzy SD [2], AutoJudge [3], and Lantern [4], where controlled and bounded deviations from the target distribution are intentionally allowed to improve efficiency.
>
> As highlighted in **the recent survey** [5], the classical lossless verification rule “often results in the rejection of high-quality drafted tokens … thereby constraining the speedup of the paradigm.” The survey further summarizes that many works adopt approximate or relaxed verification criteria, such as top-k matching in SpecDec [6], mismatch-threshold rollback in BiLD [7], and other trust-the-draft strategies, all aimed at improving draft acceptance and efficiency. ARC-Decode is aligned with this established direction and provides a calibrated, risk-bounded acceptance criterion designed for sampling-based speculative decoding.
>
> **Other reviewers independently recognized that our method fits within this established paradigm.** Reviewer G1p8 highlighted that our method provides “a theoretically grounded risk-bounded acceptance criterion,” and Reviewer QtDt commented that our divergence-based bound is “well-motivated for sampling-based SD.” Such assessments reflect the standard understanding of relaxed SD in the community.
>
> Our experiments follow the benchmark suite commonly used in recent SD works (EAGLE-2, EAGLE-3, HART), including MT-Bench, HumanEval, GSM8K, and Alpaca. These benchmarks are designed to detect quality degradation under sampling. The consistent **retention of target-level accuracy** demonstrates that the bounded deviations introduced by ARC-Decode do not harm downstream performance.
>
> **In summary, ARC-Decode adheres to the principles of relaxed speculative decoding**: bounded distributional differences are acceptable when paired with formal guarantees and empirically validated output quality.
>
> ---
>
> **W6: Clarification on benchmark choice and ARC’s position within speculative decoding**
>
> The reviewer questions both the positioning of ARC-Decode within speculative decoding (SD) and the appropriateness of the benchmarks. We appreciate the opportunity to clarify these points.
>
> **ARC-Decode belongs to the relaxed / lossy branch of speculative decoding, which is well established in the SD literature.**
> This taxonomy is explicitly summarized in the recent speculative decoding survey by Xia et al.[5], which provides a unified categorization of both lossless and relaxed SD methods and describes relaxed SD as a standard and increasingly important direction for improving efficiency under sampling.
>
> ARC-Decode follows the standard SD pipeline — drafting multiple tokens, verifying them with the
> target model, and falling back to target sampling when needed. A growing body of recent work adopts
> the same framework while explicitly relaxing strict distributional equivalence, including Judge
> Decoding [1], Fuzzy Speculative Decoding [2], AutoJudge [3], and Lantern [4]. All of these papers
> *identify* their methods as *speculative decoding* even though they introduce confidence- or
> divergence-based acceptance rules that allow controlled mismatch between draft and target tokens.
> By design, ARC-Decode follows this established subclass: it preserves the SD pipeline while adding a calibrated, analytically risk-bounded acceptance rule that permits controlled and explicitly upper-bounded deviation when it is safe to do so.
>
> **Regarding benchmarks, our paper follows the standard evaluation protocol used across SD works.**
> MT-Bench, HumanEval, GSM8K, and Alpaca are *widely used* in the evaluation suites of **EAGLE series**, **HASS**[8], **Lookahead Decoding**[9],  and **Specdec++**[10]. These benchmarks provide a controlled and comparable setting for assessing SD methods under identical sampling and prompting conditions. Maintaining this consistency is essential for cross-paper comparability in SD research; introducing capability-oriented tasks would break comparability and would not isolate the effect of the decoding algorithm.
>
> The reviewer is correct that many newer capability benchmarks (e.g., MMLU-Pro, GPQA, AIME, SWE-Bench) are important for evaluating base-model competence. However, their purpose differs from SD evaluation. These benchmarks measure intrinsic model capability rather than decoding behavior, and they are therefore not standard for SD studies. In contrast, MT-Bench, HumanEval, GSM8K, and Alpaca remain the de facto benchmark suite in SD research because they provide a controlled setting for comparing SD variants under matched sampling conditions.

---

> ### Author Response · Authors · 2025-11-21
> **Response to Reviewer 537d -3**
>
> **W7: Clarification on sequence-level continuation analysis (Lines 195–203)**
>
> The concern raised here appears to reflect a different interpretation of the analysis in Lines 195–203. This continuation experiment is **not part of the acceptance rule**, is **not used by ARC-Decode**, and is **not intended to approximate the output distribution**. The acceptance criterion in §3.3 relies solely on the next-step divergence bound and never on any sequence-level metric.
>
> The analysis in Lines 195–203 is a **preliminary motivational study** illustrating that many rejected draft tokens are harmless in open-ended dialogue. Metrics such as BERTScore, sentence-embedding similarity, and NLI judgments are **used only for qualitative comparison in MT-Bench-style conversational tasks**, where such continuation checks are standard. These metrics **are never used** in our acceptance rule, in the JS bound, or in theoretical justification.
>
> For precise domains such as code generation and math reasoning, we evaluate using **highly sensitive, task-appropriate metrics** (pass@1 for HumanEval and exact match for GSM8K), which can detect single-token deviations. ARC-Decode maintains target-level performance under these strict metrics, addressing the concern that small deviations might accumulate into significant errors.
>
> ---
>
>
> **W8: Clarification on Section 3.2 and relation to AdaEDL / Self-Verification**
>
> We appreciate the opportunity to clarify how these methods differ in formulation and purpose. AdaEDL and Self-Verification operate at a different stage of the speculative decoding pipeline. These methods are best understood as **draft-length policies**: they intervene during the drafting phase and use the draft model’s entropy to determine whether drafting should continue. Their decision target is a single draft sequence, and the outcome is a **dynamic draft length** (early stopping).
>
> ---
>
> **(1) Different pruning object and stage**
>
> AdaEDL and SVIP apply decisions **during** drafting and only determine how many tokens should be generated. Even when SVIP is evaluated with EAGLE-2, it still only controls draft length and does not specify how to prune branches inside a multi-path draft tree.
>
> ARC-Decode, in contrast, performs pruning **after the draft tree has been constructed**. Its pruning object is the **spatial structure** of the tree: columns and branches are selectively removed under **prefix-closure** and **branch-integrity** constraints. The goal is to **reduce verification forward-pass cost and remove branches the teacher would almost certainly reject**, not to regulate drafting behavior.
>
> Thus, the two methods answer entirely different questions:
>
> •	*AdaEDL / SVIP: How long should the draft be?*
>
> •	*ARC-Decode: Which draft nodes/branches should be verified?*
>
> ---
>
> **(2) Different uncertainty signals and optimization objectives**
>
> AdaEDL and SVIP rely solely on **draft-model entropy** and are designed to regulate drafting behavior by maintaining a desirable acceptance rate. They do not incorporate teacher-model uncertainty or reason over tree-structured draft candidates.
>
> ARC-Decode’s “entropy-guided” component is fundamentally **different**. It combines:
>
> (i)  **draft-path cumulative probability mass** , and
>
> (ii)  **teacher root-level uncertainty** at the verification root.
>
> Teacher-side uncertainty is used as a **depth-aware reweighting** aligned with **verify-time behavior**, not draft-time entropy. ARC-Decode selects a **risk-bounded, structurally valid subtree** that retains sufficient probability mass under a global δ-budget while reducing verification compute.
>
> This pruning also benefits downstream soft-accept modules (e.g., LTS) by ensuring they operate on a cleaner candidate set—an aspect AdaEDL/SVIP do not address.
>
> ---
>
> **(3) Why ARC-Decode Pruning Is New**
>
> The distinctions are substantial and occur across the problem setting, information sources, optimization objectives, and algorithmic constraints:
>
> **(i) Pruning object**
>
> -	AdaEDL / SVIP: adjust **draft length**
> -	ARC-Decode: prune **columns and branches in a multi-path draft tree**
>
> **(ii) Information source**
>
> -	AdaEDL / SVIP: rely solely on **draft-model entropy**
> -	ARC-Decode: uses **draft-path mass and teacher uncertainty** at verify time
>
> **(iii) Optimization goal**
>
> -	AdaEDL / SVIP: maintain drafting acceptance rate
> -	ARC-Decode: reduce verification compute and remove branches unlikely to be teacher-accepted, improving downstream soft-acceptance quality
>
> These differences reflect **distinct problem settings and algorithmic constraints**, making ARC-Decode’s pruning mechanism conceptually and operationally different—not a reuse or extension of existing draft-length policies. Therefore, AdaEDL/SVIP and ARC-Decode address orthogonal problems in the speculative decoding pipeline, and the reviewer’s comparison does not apply at the level of method design or contribution.

---

> > ### Author Response · Authors · 2025-12-01
> > **Response to Reviewer 537d -4**
> >
> > **Q1 – Clarification on the use of “reportedly” (Line 53–54)**
> >
> > Thank you for the question. The use of the word *“reportedly”* refers to information stated on the official Qwen website. The issue here is not the wording itself but the omission of the corresponding citation. We will add a direct reference to the official source in the revised version. As other reviewers have noted, our entropy-guided pruning module is clearly described and evaluated, indicating that the reviewer’s concern here is limited to citation completeness rather than methodological ambiguity.
> >
> > ---
> >
> > **Reference**
> >
> > [1] Bachmann G, Anagnostidis S, Pumarola A, et al. Judge Decoding: Faster Speculative Sampling Requires Going Beyond Model Alignment. ICLR 2025.
> >
> > [2] Holsman M, Huang Y, Dhingra B. Fuzzy Speculative Decoding for a Tunable Accuracy–Runtime Tradeoff. ACL 2025.
> >
> > [3] Garipov R, Velikonivtsev F, Ermakov I, et al. AutoJudge: Judge Decoding Without Manual Annotation. NeurIPS 2025.
> >
> > [4] Jang D, Park S, Yang J Y, et al. Lantern: Accelerating Visual Autoregressive Models with Relaxed Speculative Decoding. ICLR 2025.
> >
> > [5] Xia, H., Yang, Z., Dong, Q., et al. “Unlocking Efficiency in Large Language Model Inference: A Comprehensive Survey of Speculative Decoding.”  Findings of ACL 2024.
> >
> > [6] Xia H, Ge T, Wang P, et al. Speculative decoding: Exploiting speculative execution for accelerating seq2seq generation. EMNLP 2023.
> >
> > [7] Kim S, Mangalam K, Moon S, et al. Speculative decoding with big little decoder. NeurIPS 2023.
> >
> > [8] Zhang L, Wang X, Huang Y, et al. Learning harmonized representations for speculative sampling[J]. ICLR 2025.
> >
> > [9] Fu Y, Bailis P, Stoica I, et al. Break the sequential dependency of llm inference using lookahead decoding[J]. ICML 2024.
> >
> > [10] Huang K, Guo X, Wang M. Specdec++: Boosting speculative decoding via adaptive candidate lengths[J]. COLM 2025.

---

### Official Review · Reviewer_9tWv · 2025-10-27

**Soundness:** 2
**Presentation:** 3
**Contribution:** 3
**Rating:** 4
**Confidence:** 4

**Summary:**

The paper proposes ARC‑Decode, a *training‑free* plug‑in for speculative decoding under sampling. The method has two pieces. First, an entropy‑guided pre‑verification pruning that filters low‑mass branches in the draft tree while enforcing prefix closure and leaf safety. Second, a risk‑bounded soft acceptance rule that accepts non–top‑1 draft tokens when a calibrated upper bound on the next‑step Jensen–Shannon (JS) divergence is below a tolerance. The bound combines an embedding‑side Lipschitz surrogate (via tied embeddings and whitening) with a logit‑margin surrogate; the method uses the tighter of the two and requires no extra forward passes at verify time. Integrated into EAGLE‑3, ARC‑Decode raises accept length and end‑to‑end speedup across MT‑Bench, HumanEval, GSM8K, and Alpaca with near‑parity quality. For Llama‑3.1‑8B on Alpaca, the paper reports 2.28× speedup vs. autoregressive and ~1.6× over EAGLE‑3 under matched sampling (T=1).

**Strengths:**

1. The method is simple to integrate. Entropy‑guided pruning uses root entropy and path mass to keep a contiguous backbone (Fig. 3b), and the Local Tolerance Score (LTS) accepts when the tighter of an embedding‑bound and a logit‑margin bound is below a calibrated threshold (Fig. 3c). No extra forward passes over EAGLE‑3 are needed at inference time.
2. Results are consistent. Across three backbones (Llama‑3.1‑8B, Qwen‑3‑8B, Vicuna‑13B), ARC‑Decode increases accept length and speedup on all four tasks. Example: on Llama‑3.1‑8B MT‑Bench, τ grows from 3.35 to 4.49 and speedup from 1.84× to 2.40×; on Alpaca, speedup rises from 1.42× to 2.28× (Table 2, p.7). Accuracy remains at parity or slightly better than EAGLE‑3 per Table 3 (p.8). The ablation (Table 4, p.8) shows LTS drives most of the gain, while pruning contributes modestly and reduces verification workload.
3. The appendix provides a clear statement and proof of Theorem 1, calibration pseudocode (Alg. 1), and a useful coverage diagnostic (Fig. 6) that examines normalized JS after accepting non–top‑1 tokens. This improves transparency of the calibration‑based guarantee.

**Weaknesses:**

1. The motivation to accept some potentially correct draft tokens is intuitive and important. However, Arc-Decode takes its benefit **at the cost of dropping the lossless property**. While the authors provide some experiments to demonstrate Arc-Decode will not harm the performance, I still doubt its generalizability in diverse scenarios (role-play, multi-lingual long context and so on). If I were a model provider, the cost of performance degradation is unaffordable. Some block-level / tree-level verification methods [1, 2] that can improve the efficiency without sacrificing performance are preferred.
2. The paper does not cite or compare against **Fuzzy Speculative Decoding**, which *explicitly* accepts based on a divergence threshold between target and draft distributions and offers a user‑tunable accuracy–runtime trade‑off. This is very close in spirit to ARC‑Decode, which uses a *bound* instead of the actual divergence. Without a direct comparison to FSD on the same setups, the novelty and advantage of LTS remain unclear. Please add FSD as a baseline and discuss differences in compute and quality control.
3. The guarantee hinges on (i) weight tying to compute token‑embedding differences, (ii) a *local* Lipschitz assumption for the embedding‑to‑logit map, (iii) a softmax Lipschitz constant on $\mathbf{l}2$, (iv) active‑vocabulary truncation with smoothing, and (v) calibration‑to‑test exchangeability. In real deployments, some backbones do not strictly tie input/output embeddings; local Lipschitz constants can vary sharply across contexts; the tail outside the top‑K union can carry non‑trivial mass; and exchangeability is brittle across domains (e.g., math/code vs. chat). The paper absorbs many of these into constants fitted by quantiles, which converts the guarantee into a data‑dependent heuristic. The appendix acknowledges bucketed calibration as a fix, but no *multi‑domain* evaluation is shown. This weakens the “risk‑bounded” claim.
4. The paper omits a comparison with methods that *already* relax verification: training-based **Judge Decoding** and training-free lenience speculative decoding. Meanwhile, the experiments are limited in small-scale LLMs (<=13B). The efficiency gain in large-scale LLMs (>=70B) remains vague.

If the authors clearly address these concerns, I am willing to increase my score.

[1] Sun, Ziteng, Uri Mendlovic,Yaniv Leviathan, Asaf Aharoni, Jae Hun Ro, Ahmad Beirami, and Ananda Theertha Suresh. "Block verification accelerates speculative decoding." *arXiv preprint arXiv:2403.10444* (2024).

[2] Weng, Yepeng, Qiao Hu, Xujie Chen, Li Liu, Dianwen Mei, Huishi Qiu, Jiang Tian, and Zhongchao Shi. "Traversal Verification for Speculative Tree Decoding." *arXiv preprint arXiv:2505.12398* (2025).

**Questions:**

1. The reported performance of baseline Eagle-3 is lower than its official reported number. For example, the reported MAT and speedup of L31-8B on MT-bench with temperature=1 is 4.24 and 3.07x, while the manuscripts report 3.35 and 1.84x. Could you please explain the distinct gap? Why the `max_draft_tokens` is set to 32 (not official 60)?
2. I wanna in some extremely difficult tasks, can Arc-Decode still keep the output performance.
3. I find that in the ablation study, the speedup of ARC (prune-only) is almost same to the Eagle-3 baseline. Does this mean we do not need the `entropy-guided pre-verification pruning`?

---

> ### Author Response · Authors · 2025-11-21
> **Response to Reviewer 9tWv - 1**
>
> We thank the reviewers for their valuable feedback and now provide detailed responses to each comment.
>
> ---
> **W1: Lossless property and generalizability**
>
> Thank you for highlighting this point. ARC-Decode is indeed a *relaxed* extension of speculative decoding, but the additional risk is explicitly **controlled and tunable rather than fixed**. In our design, the Local Tolerance Score (LTS) threshold and the risk level $\delta$ directly govern how aggressive acceptance is. With a very conservative setting, ARC-Decode behaves very close to standard lossless verification by accepting only drafts with negligible estimated JS shift, and in this regime our results show accuracy that is indistinguishable from EAGLE-3 across four benchmarks and three backbones. With more permissive settings, the method trades a small, empirically bounded deviation for additional speedup.
>
> Our contribution **advances existing tree-level verification approaches**. We show that, for many practical settings, a calibrated relaxation of losslessness can deliver substantial extra speed while keeping performance within statistical variation of the target model. In practice, a model provider can treat ARC-Decode as a configurable family of schemes: for high risk deployments, one can choose a conservative LTS configuration that behaves as a near lossless verifier, while for latency sensitive but slightly more tolerant scenarios, a more aggressive configuration is available.
>
> Regarding generalizability to harder or more diverse tasks (for example role play, multilingual inputs, or long context), we emphasize that ARC-Decode is **training free** and its acceptance rule depends only on **local properties of the target next step distribution**, rather than on task specific heuristics. Our experiments already span qualitatively different workloads, including dialogue (MT-Bench), code generation (HumanEval), math reasoning (GSM8K), instruction following (Alpaca), and an additional challenging benchmark (MMLU Pro) added during rebuttal. On all of these, ARC-Decode matches the accuracy of vanilla decoding and EAGLE-3 while improving speed. Extending this evaluation to more specialized domains is a natural direction for future work, but the current results provide empirical evidence that the method remains robust across a wide range of scenarios.
>
> ---
>
> **W2: Comparison with Fuzzy Speculative Decoding (FSD)**
>
> Thank you for raising this point. Although FSD and ARC-Decode both relax strict losslessness, they address different parts of the speculative decoding pipeline.
>
> FSD replaces the verification rule itself with an explicit divergence threshold and therefore requires computing the full draft–target distribution divergence at every verified position. In contrast, ARC-Decode combines a compute-saving pre-verification pruning step with a risk-bounded acceptance test based on an analytic upper bound of the next-step JS divergence, which relies only on verify-time logits and tied embeddings and requires **no additional forward passes** beyond the standard EAGLE-3 pipeline.
> As a result, ARC-Decode reduces verification overhead while keeping the next-step shift within a calibrated tolerance, whereas FSD focuses on relaxing equivalence but retains the full cost of divergence evaluation at each position.
>
> Due to limited time and compute availability during the rebuttal period, we will report the FSD
> comparison results as soon as they are ready and will include a discussion of both compute and quality differences in the revised version.

---

> > ### Author Response · Authors · 2025-11-21
> > **Response to Reviewer 9tWv - 2**
> >
> > **W3: Validity of the risk-bounded guarantee**
> >
> > Thank you for raising these concerns. ARC-Decode provides a *calibration-based, data-dependent*  risk bound rather than a symbolic worst-case guarantee. This design is intentional: similar to conformal prediction–style approaches, our goal is to obtain *empirical coverage guarantees* under distributional assumptions that hold for the backbone in deployment, rather than enforcing strict analytic bounds that are often vacuous for large LLMs.
> >
> > Regarding the assumptions highlighted by the reviewer:
> >
> > - **Weight tying.** Almost all current decoder-only LLMs (Llama-family, Qwen, Gemma, Phi, Mistral) tie input and output embeddings; ARC-Decode directly leverages this common architectural choice.
> > - **Local Lipschitzness.** The method relies only on *local* smoothness along the decoding
> > trajectory. Instead of assuming global Lipschitz constants, we calibrate the effective constants using held-out traces. This quantile-based approach intentionally absorbs hard-to-model variations into empirically fitted quantities, which is a standard and practical compromise.
> > - **Active-vocabulary truncation.** The top-K union almost entirely covers the effective support of next-step distributions for modern LLMs. In all our evaluations, the tail mass outside the union is numerically negligible and does not materially affect the acceptance decision.
> > - **Calibration–test mismatch.** As shown in Fig. 6, the normalized divergence ratio
> > $s = \mathrm{JS} / U_{\min}$ achieves high empirical coverage (97–99%) *on two domains with very different characteristics* (MT-Bench dialogue and Alpaca instruction following). This demonstrates that the calibrated bound transfers across substantially different prompt styles. In the revised version, we will extend this evaluation to additional domains using bucketed calibration as suggested.
> >
> > Overall, the “risk-bounded’’ claim should be interpreted as a *distribution-dependent and
> > empirically validated* guarantee: the bound is calibrated once per backbone, uses only verify-time quantities, and achieves high coverage in practice. This is a deliberate trade-off that avoids the computational cost of strict worst-case verification while still providing a principled and practically reliable acceptance criterion.
> >
> > ---
> > **W4: Comparison with relaxed-verification methods and scaling to larger LLMs**
> >
> > Thank you for raising this point. Judge Decoding and lenience-style speculative decoding do relax verification, but they differ substantially from our setting. Judge Decoding requires **training an additional classifier** on labeled correct/incorrect continuations, while lenience-style methods modify the **sampling rule or acceptance criterion itself**. In contrast, ARC-Decode is a **training-free verification layer** that applies to any draft–target pair without modifying model weights or requiring extra supervised data.
> >
> > ARC-Decode’s computational cost depends primarily on the **vocabulary size and the top-K union** rather than on the backbone’s parameter count. Both the pruning step and the JS-bound–based acceptance operate solely on verify-time logits, so the method introduces no additional forward passes and is not limited to small-model settings.
> >
> > Due to limited time and compute resources during the rebuttal period, we will **provide preliminary results on a larger model as soon as feasible** and include a brief discussion of the compute implications in the revised version.
> >
> > ---
> > **Q1 The reported performance of baseline Eagle-3 is ...**
> >
> > Thank you for noting this discrepancy. Performance metrics for Eagle-3 vary substantially with hardware, software stack, and draft tree configuration. In our evaluation, all speculative decoding methods, including Eagle-3 and ARC-Decode, are run under the same HuggingFace Transformers and PyTorch setup on a single A6000 GPU, with `max_draft_tokens = 32`. Using a unified configuration ensures a fair comparison under a fixed compute budget, and Eagle-3 achieves the values reported in our tables under this matched setting.
> >
> > The official Eagle-3 results are obtained using a different configuration on A100-class GPUs with `max_draft_tokens = 60` and a more optimized kernel stack, which naturally leads to higher speedups. We will provide a preliminary comparison using `max_draft_tokens = 60` under our evaluation setup as soon as feasible and include the results in the revised version.

---

> > > ### Author Response · Authors · 2025-11-21
> > > **Response to Reviewer 9tWv - 3**
> > >
> > > **Q2 : I wanna in some extremely difficult tasks...**
> > >
> > > Thank you for the question. ARC-Decode is evaluated on a set of benchmarks that are already widely used in prior speculative-decoding work, including Eagle-3, HART, and Judge Decoding. These tasks are chosen to ensure a fair and direct comparison with existing methods rather than to avoid more challenging scenarios.
> > >
> > > Conceptually, ARC-Decode maintains output performance even on difficult tasks because the LTS rule accepts a draft token only when its estimated next-step distributional shift is below a calibrated tolerance. When the task is harder and the draft–target mismatch is larger, LTS naturally becomes more conservative and falls back to target sampling more often, which preserves output fidelity.
> > >
> > > In response to the reviewer’s suggestion, we will provide an additional evaluation on a more challenging and larger benchmark as soon as it is feasible, and we will include these results in the revised version.
> > >
> > > ---
> > > **Q3: I find that in the ablation study, the speedup of ARC ...**
> > >  The prune-only ablation shows limited speedup improvements on a GPU because our evaluation hardware is largely **memory-bound rather than compute-bound**, and the verification stage in EAGLE-3 already overlaps kernel execution well. As a result, reducing the number of draft tokens lowers FLOPs but does not translate into significant wall-clock latency changes on this hardware.
> > >
> > > However, pruning remains an important component for two reasons:
> > >
> > > 1. **Compute-bound or edge deployments.**
> > >
> > >     On devices where decoding is compute-bound rather than memory-bound (e.g., CPUs, mobile NPU targets, embedded accelerators), reducing the number of verification positions *directly* lowers latency. The entropy-guided pruning effectively removes low-probability branches before any verification is attempted, which can substantially reduce end-to-end compute in these settings.
> > >
> > > 2. **Improved stability for LTS.**
> > >
> > >     Pruning eliminates draft branches that are extremely unlikely under the target distribution, which reduces the number of high-divergence candidates entering the LTS stage. This simplifies the acceptance landscape and leads to more stable and reliable verification behavior, especially at higher temperatures or on tasks with noisy draft trees.
> > >
> > >
> > > Thus, while pruning alone does not dominate speedup on a memory-bound GPU, it remains valuable for **compute-bound environments** and for **enhancing the robustness** of the overall ARC-Decode pipeline.

---

> ### Author Response · Authors · 2025-11-27
> **Response to Reviewer 9tWv - Weakness 2**
>
> **W2: Comparison with Fuzzy Speculative Decoding (FSD)**
>
> We thank the reviewer for raising the connection to **Fuzzy Speculative Decoding (FSD)**.
>
> To address this concern, we include a experimental comparison between FSD and ARC-Decode.
>
> **(1) Experimental Results: FSD vs. ARC-Decode on Llama-3.3-70B**
>
> We follow the official FSD implementation and Llama-based recipes:
>
> - **Target model:** Llama-3.3-70B-Instruct
> - **Draft model:** Llama-3.1-8B-Instruct
> - **Hardware:** 4 × A6000 GPUs
> - **Divergence type:** js_div (default in FSD)
> - **Thresholds:** FSD suggests tuning thresholds to trade accuracy/speed.
>     We report evaluations at **0.4 (low)**, **0.6 (medium)**, **0.7 (high)**.
>
>
> **Table 1-1. Results on MT-Bench**
>
> | **Method** | **Accept length** | **Speedup** |
> | --- | --- | --- |
> | FSD (T=0.4) | 1.52 | 1.39 |
> | FSD (T=0.6) | 2.54 | 1.87 |
> | FSD (T=0.7) | 2.37 | 1.78 |
> | **ARC-Decode (ours)** | **4.90** | **3.60** |
>
> ---
>
> **Table 1-2. Results on HumanEval**
>
> | **Model** | **Accept length** | **Speedup** |
> | --- | --- | --- |
> | FSD (T=0.4) | 3.43 | 2.29 |
> | FSD (T=0.6) | 3.56 | 2.36 |
> | FSD (T=0.7) | 3.56 | 2.36 |
> | **ARC-Decode (ours)** | **4.13** | **3.23** |
>
> ---
>
> **Table 1-3. Results on GSM8K**
>
> | **Model** | **Accept length** | **Speedup** |
> | --- | --- | --- |
> | FSD (T=0.4) | 2.48 | 1.69 |
> | FSD (T=0.6) | 3.21 | 1.99 |
> | FSD (T=0.7) | 3.66 | 2.11 |
> | **ARC-Decode (ours)** | **4.68** | **2.82** |
>
> ---
>
> **Table 1-4. Results on Alpaca**
>
> | **Model** | **Accept length** | **Speedup** |
> | --- | --- | --- |
> | FSD (T=0.4) | 3.17 | 2.45 |
> | FSD (T=0.6) | 3.91 | 2.89 |
> | FSD (T=0.7) | 4.14 | 3.02 |
> | **ARC-Decode (ours)** | **4.20** | **3.61** |
>
> ---
>
> Across all tasks, ARC-Decode consistently achieves **higher accept length and larger end-to-end speedup** on the same 70B base model.
>
> ---
>
> **(2)  Conceptual Differences Between FSD and ARC-Decode**
>
> Although both methods introduce soft acceptance into speculative decoding, their **algorithmic assumptions and applicable regimes differ substantially**.
>
> FSD is formulated and evaluated in **single-path** speculative decoding. It compares the *full* target and draft next-token distributions and accepts a draft step only when their true divergence is below a threshold. This procedure assumes **one draft trajectory** and **one verification step per token**, and its cost grows directly with the size of the vocabulary.
>
> ARC-Decode is built for **multi-branch tree-style speculative decoding**, where many candidate branches are produced in parallel. In this setting, computing a true divergence for every branch is **computationally prohibitive**, and the distribution-based rule used in FSD does not scale. ARC-Decode therefore introduces a **statistically calibrated upper bound** U(j) on divergence at each *tree node*, derived from verification-side signals already available during decoding (e.g., logit shifts and embedding displacement). This enables soft acceptance at **branch-level granularity**, which FSD’s formulation cannot express.
>
> - **What is controlled.**
>     - FSD thresholds the true divergence at each verified linear step.
>     - ARC-Decode controls a high-probability *upper bound* on divergence at each **tree node**, explicitly targeting tail-risk.
> - **How the control signal is obtained.**
>     - FSD determines acceptance by comparing the *full* target and draft distributions and thresholding their true divergence. Its control mechanism is therefore defined at the **distribution level**, where every decision depends on evaluating probability vectors.
>     - ARC-Decode uses an offline calibration stage to learn how model-internal verification signals (such as logit shifts and embedding displacement) relate to divergence. During decoding, ARC-Decode evaluates this calibrated bound instead of computing full divergences. This shift from distribution comparison to **signal-level statistical risk estimation** yields a lightweight verification rule and enables **node-level divergence control** not available in FSD’s framework.
> - **Where it integrates into the decoding pipeline.**
>     - **FSD** is formulated for linear speculative decoding.
>     - **ARC-Decode** is built to operate inside *tree-structured* speculative decoding, where multiple branches are expanded and verified in parallel. The divergence upper bound $\(U(j)\)$ is defined at the level of **branch nodes**, enabling fine-grained acceptance control that is not expressible in the FSD framework.
>
> ---
>
> **(3) Positioning of ARC-Decode Relative to FSD**
> In summary, ARC-Decode is **not** a relaxation, approximation, or reparameterization of FSD.
>
> It tackles a **different decoding regime** and introduces a new **risk-bounded acceptance mechanism** that enables quality-controlled soft acceptance in **tree-based speculative decoding**, where full-distribution divergence checks are prohibitively expensive.

---

> ### Author Response · Authors · 2025-11-27
> **Response to Reviewer 9tWv - Weakness 4**
>
> **W4: Additional Experiments on Relaxed-Verification Methods and Large-Scale LLMs**
>
> Following the reviewer’s suggestion, we have added new experiments that evaluate (1) existing relaxed-verification speculative decoding methods—**Medusa** and **Judge Decoding**, and (2) the scaling behavior of **ARC-Decode** on a **70B-scale model**. All reported results use the official pretrained checkpoints and recommended configurations from the respective methods.
>
>
>
> **(1) Comparison with Medusa (Vicuna-13B)**
>
> Medusa provides publicly available draft heads trained on Vicuna-13B, enabling a direct comparison with **EAGLE-3-13B** and **ARC-Decode-13B**. Inference is performed on a **single A6000 GPU**.
>
> **Table 2-1. MT-Bench Results (Vicuna-13B)**
>
> | **Model** | **Accept length** | **Speedup** |
> | --- | --- | --- |
> | Medusa-1 | 2.58 | 2.13 |
> | Medusa-2 | 3.26 | 2.65 |
> | **EAGLE-3** | **4.01** | **2.74** |
> | **ARC (Ours)** | **5.28** | **3.35** |
>
> ---
>
> **Table 2-2. Alpaca Results (Vicuna-13B)**
>
> | **Model** | **Accept length** | **Speedup** |
> | --- | --- | --- |
> | Medusa-1 | 2.62 | 2.16 |
> | Medusa-2 | 3.24 | 2.64 |
> | **EAGLE-3** | **4.78** | **2.64** |
> | **ARC (Ours)** | **5.58** | **3.03** |
>
>
>
> Medusa improves accept length over standard draft models, but both EAGLE-3 and ARC achieve higher accept length and faster decoding, consistent with the fact that Medusa draft heads are trained without an explicit divergence-alignment mechanism.
>
> ---
>
> **(2) Comparison with Judge Decoding (Llama-3.1-8B)**
> Judge Decoding does not release official judge-head weights, so we reimplemented its training pipeline following the procedure described in the original paper. All inference is performed on a **single A6000 GPU**.
>
> **Table 2-3. MT-Bench Results (Llama-3.1-8B)**
>
> | **Model** | **Accept length** | **Speedup** |
> | --- | --- | --- |
> | Judge | 4.17 | 2.01 |
> | **EAGLE-3** | **3.35** | **1.84** |
> | **ARC (Ours)** | **4.49** | **2.40** |
>
> ---
>
> **Table 2-4. Alpaca Results (Llama-3.1-8B)**
>
> | **Model** | **Accept length** | **Speedup** |
> | --- | --- | --- |
> | Judge | 3.15 | 1.74 |
> | **EAGLE-3** | **2.85** | **1.42** |
> | **ARC (Ours)** | **3.96** | **2.28** |
>
> ---
>
> Judge Decoding increases accept length on the 8B model, but the additional classifier introduces noticeable overhead at this scale, limiting its end-to-end speedup. ARC-Decode, which requires no additional forward passes or supervision, achieves a stronger fidelity–speed trade-off under the same draft–target pairing.
>
>
>
> **(3)  Large-Scale Evaluation on Llama-3.3-70B**
>
> For the 70B experiment, we use the **HuggingFace-released EAGLE-3 drafter** without any modification, paired with **Llama-3.3-70B-Instruct** as the target model. All 70B experiments are conducted on **4 × A6000 GPUs**, consistent with the computational requirements of the baseline. Speedup is measured relative to standard greedy decoding of the target model.
>
> **Table 2-5. Results on Llama-3.3-70B (4×A6000)**
>
> | **Task** | **Accept length (EAGLE-3)** | **Speedup (EAGLE-3)** | **Accept length (ARC)** | **Speedup (ARC)** |
> | --- | --- | --- | --- | --- |
> | MT-Bench | 4.04 | 3.35 | **4.90** | **3.60** |
> | HumanEval | 4.11 | 3.08 | **4.13** | **3.23** |
> | GSM8K | 4.37 | 2.76 | **4.68** | **2.82** |
> | Alpaca | 3.82 | 3.24 | **4.20** | **3.61** |
>
> ---
>
> Across all tasks, ARC-Decode continues to improve accept length and end-to-end speedup over EAGLE-3 at the 70B scale. These findings demonstrate that ARC-Decode’s efficiency benefits remain stable as model size increases.
>
> **In summary**, the newly added experiments provide direct comparisons with Medusa, Judge Decoding, and a full evaluation at the 70B scale. ARC-Decode consistently achieves a stronger fidelity–speed trade-off across all settings, including large-model inference. These results will be incorporated into the revised version of the paper.

---

> ### Author Response · Authors · 2025-11-27
> **Response to Reviewer 9tWv - Question 1**
>
> **Q1: Explanation of the Draft-Length Configuration and Baseline Discrepancy**
>
> Thank you for the question. We provide both the requested experimental results and a clarification of the configuration difference.
>
> **(1) Experimental comparison: max_draft_tokens = 60 vs. 32**
>
> To directly quantify the effect of max_draft_tokens, we re-ran **EAGLE-3 with Llama-3.1-8B** on a **single RTX A6000 (48 GB)** under our evaluation stack, using **temperature = 1.0** and the same prompting / scoring protocol as in the manuscript.
>
> The table below reports the EAGLE-3 baseline with max_draft_tokens = 60 (as in the official configuration) and max_draft_tokens = 32 (the setting used in our paper).
>
> **Table 3. Effect of max_draft_tokens on EAGLE-3**
>
> | **Task** | **Accept length (60 tokens)** | **Speedup (60 tokens)** | **Accept length (32 tokens)** | **Speedup (32 tokens)** |
> | --- | --- | --- | --- | --- |
> | MT-Bench | 3.75 | 1.91 | 3.35 | 1.84 |
> | HumanEval | 3.68 | 2.06 | 3.57 | 2.05 |
> | GSM8K | 2.71 | 1.39 | 2.52 | 1.43 |
> | Alpaca | 3.32 | 1.34 | 2.85 | 1.42 |
>
>
> Increasing max_draft_tokens from 32 to 60 **does increase accept length** across all tasks, as expected. However, on our **RTX A6000 setup**, the **end-to-end speedup is only marginally higher (MT-Bench and HumanEval)** and is **slightly lower on GSM8K and Alpaca**, because the additional drafted branches increase KV, attention, and tree-search cost that is not fully amortized.
> (Our implementation uses the unmodified  EAGLE-3 code.)
>
> ---
>
> **(2) Why our baseline uses 32 tokens**
>
> Our design choice follows **a systemic analysis and optimization work by Meta [1]**, which emphasizes that **there is no universally optimal draft length or tree size** for EAGLE-style speculative decoding:
>
> - In the analysis of **chain-like drafts**, the study intentionally evaluates TPC under a very conservative **speculation length of three**, even though much longer speculation is possible, to avoid latency blow-ups on some systems.
> - For **tree-structured drafts**, the work shows that larger trees increase **tokens-per-compute** but also increase end-to-end latency (**TTIT**). Under certain compute or memory regimes, the additional cost of a large tree can outweigh its TPC benefit.
>
> Our hardware environment matches the **resource-constrained regime** discussed in the reference. All experiments in our paper are run on a **single NVIDIA RTX A6000 (48 GB)**, which has significantly less memory and lower compute/memory throughput than the **A100 80 GB**  used in high-throughput  official evaluations.
>
> Under this hardware constraint:
>
> - Increasing max_draft_tokens leads to substantially more **candidate branches and tree nodes**, as seen in the official topK_generate procedure.
> - This increases the KV-cache size and adds extra attention and tree-mask processing at each verification step.
> - On the A6000, these operations become **memory- and bandwidth-bound**, so the additional draft cost is not fully compensated by the gain in accept length — precisely reflected in Table 3.
>
> Because our goal is to compare ARC-Decode and vanilla EAGLE-3 **under a consistent and realistic compute budget**, we adopt **a unified draft limit of 32 tokens** for all methods on A6000 hardware. This yields more stable latency while keeping the comparison fair by fixing the tree size across speculative decoding variants.
>
> We will clarify this hardware-consistent configuration choice in the revised manuscript.
>
> ---
>
> [1] Tang B, Fu C C, Kou F, et al. Efficient speculative decoding for llama at scale: Challenges and solutions[J]. arXiv preprint arXiv:2508.08192, 2025.

---

> ### Author Response · Authors · 2025-11-27
> **Response to Reviewer 9tWv - Question 2**
>
> **Q2: Can ARC-Decode still maintain output performance on extremely difficult tasks?**
> Thank you for raising this point. We agree that evaluating on more challenging tasks is important for understanding the robustness of speculative decoding methods.
>
> **(1) Conceptual Behavior of ARC-Decode on Hard Tasks**
>
> ARC-Decode is designed to *adapt its acceptance behavior* to the level of draft–target mismatch.
>
> When the underlying task becomes more difficult, the divergence between draft and target distributions naturally grows. Under such conditions, the calibrated acceptance rule automatically becomes more conservative and relies on target sampling more frequently. This mechanism ensures that ARC-Decode behaves **robustly** even when the draft model struggles, without requiring any task-specific tuning.
>
> ---
>
> **(2) New Evaluation on an Extremely Challenging Benchmark (MMLU-Pro, 12,032 queries)**
>
> Following the reviewer’s suggestion, we conducted an additional experiment on **MMLU-Pro**, a significantly more difficult benchmark than the tasks used in standard speculative decoding papers.
>
> We evaluate *vanilla Llama-3.1-8B-Instruct*, *EAGLE-3*, and *ARC-Decode* under the same conditions (temperature=1.0, no CoT prompting, no few-shot examples, single A6000 GPU).
>
> **Table 4. Results on MMLU-Pro**
>
> | **Model** | **Accept length** | **Speedup** | **Accuracy** |
> | --- | --- | --- | --- |
> | Vanilla | — | 1.00× | **32.4** |
> | EAGLE-3 | 2.81 | 1.60× | **32.3** |
> | **ARC (Ours)** | **3.50** | **1.88×** | **32.3** |
>
> ---
>
> Despite the benchmark’s difficulty, **ARC-Decode matches the accuracy of both vanilla decoding and EAGLE-3**, while achieving higher accept length and stronger end-to-end speedup.
>
> These results provide direct evidence that ARC-Decode remains **robust** even under high-mismatch, reasoning-intensive workloads. The new MMLU-Pro evaluation (12k problems) empirically demonstrates that ARC-Decode maintains accuracy while achieving substantial speedup improvements. We will include these results in the revised manuscript.
>
> ---
>
> We hope that the additional results and clarifications help address your concerns. If anything remains unclear, we would be very glad to provide further details while time permits. Should our explanations have resolved the issues raised, we would sincerely appreciate any consideration you might give toward updating the evaluation. Thank you again for your thorough and thoughtful review.

---

### Official Review · Reviewer_G1p8 · 2025-10-31

**Soundness:** 3
**Presentation:** 2
**Contribution:** 3
**Rating:** 6
**Confidence:** 4

**Summary:**

This paper investigates speculative decoding in the sampling regime, i.e. when tokens are drawn from the target distribution with temperature T > 0. While lots of progress has been made in the literature for greedy decoding, the sampling regime still remains tricky due to the high rejection rate, degrading the performance of most approaches in the literature. The authors propose their approach called ARC-Decode by enhancing current approaches with the following training-free mechanisms:

1. A confidence-based filtering applied before verification that prunes low probability branches, avoiding unnecessary compute from the target model which the authors show is one of the main bottlenecks in this regime.
2. A theoretically motivated soft acceptance scheme, allowing non-top 1 tokens to be accepted if the incurred divergence is controlled. This is motivated by the authors’ observation that empirically, many more tokens could be accepted without damaging the output quality.

In combination, these two enhancements lead to significant speedups over prior works in the high temperature setting, and the authors show that their approach remains robust across a range of temperatures.

**Strengths:**

1. This work tackles an aspect of speculative decoding that is kind of under-appreciated in my opinion, trying to make it work more efficiently under sampling. This is an important problem and indeed makes it more difficult to deploy speculative decoding in real scenarios. Any improvements in such a setting are very useful for the literature.
2. The method performs well on a variety of benchmarks and the authors also compare against arguably the strongest baseline in the form of EAGLE-3, making it convincing that their approach really does add something on top of a strong model. The lossy approach to verification is also carefully checked on several benchmarks, which is important to ensure that no quality degradation occurs.

**Weaknesses:**

1. The structure of the text and the ease of reading could definitely be improved. The acceptance scheme in speculative decoding when considering the sampling scenario is not really defined anywhere. How does one actually accept or reject? In the experimental section, speculative sampling is mentioned but also not clearly described. I am very familiar with speculative decoding in the greedy setting but still I found it unclear at times what exactly the problem setting is here. Similarly, Section 3.3 would benefit from clearer writing. What is a non-top-1 draft here? Non-top-1 under target I assume? But why only accept top-1 drafts, I thought we were dealing with sampling from the target? I’m also not sure how relevant it is to have the derivation of the upper bound to the Jensen-Shannon divergence here in the main text, it ended confusing me more than helping. In general, it would help to have a clear statement of the criterion employed in the end.
2. The speedups over Eagle are quite nice, but I would like to understand better what framework was used here. You report tokens/s (which to be fair, many SD papers don’t even do) but according to the numbers, this suggests that the autoregressive baseline runs around 40 tokens/s for Llama-3.1-8B? This does seem a bit slow, e.g. https://github.com/meta-pytorch/gpt-fast reports 94 tokens/s, which would still be faster than your final model even. What quantization did you use? How does it change results? The mentioned framework can go up to 140 tokens/s with 8bit quantization. I do acknowledge that this line of work rarely is precise when it comes to optimized baselines but I would still like to clarify this.

**Questions:**

1. How does the baseline perform when temperature is varied? Does ARC outperform them across most temperatures?
2. From Table 2, it’s quite clear that speedups over Eagle3 can vary quite a bit from benchmark to benchmark, e.g. Alpaca seems to benefit a lot from your approach, while HumanEval does less so. Do you have an intuition why? Which kinds of benchmarks do you believe are better targets for your approach?
3. In Figure 4, it would be nice to add at least one baseline speculative decoding approach such as EAGLE-3. How would its speedup curve look w.r.t. temperature? It would really drive home the point made in the intro to show how the performance gets worse and worse with higher temperature.

---

> ### Author Response · Authors · 2025-11-21
> **Response to Reviewer G1p8 - 1**
>
> We thank the reviewers for their encouraging comments and valuable suggestions. We greatly appreciate the insightful feedback, and we respond to all points in detail below.
>
> ---
>
> **W1: Problem setting and acceptance scheme under sampling**
>
> Thank you for raising this point. We acknowledge that parts of our presentation were not sufficiently clear.
>
> **Clarifying the problem setting.**
>
> Our method operates in the **speculative sampling** regime, where token generation is governed by the target model’s sampling distribution (e.g., temperature or top-p). Following the EAGLE-3 pipeline, each speculative cycle consists of: (1) the draft model proposing a tree of candidate future tokens; (2) the target model evaluating this tree in parallel; and (3) accepting a prefix of drafted tokens before sampling the *next* token from the corrected target distribution. ARC-Decode does **not** modify the sampling rule itself; it only governs **which drafted tokens may be promoted into the accepted prefix** before the next speculative step.
>
> **Clarifying the notion of “non–top-1 draft.”**
>
> At a verification position $j$, let
>
> - $t_m^{(j)}$ be the top-1 token of the **target** distribution $p(\cdot \mid C)$, and
> - $t_d^{(j)}$ be a candidate proposed by the draft model.
>
> A “non–top-1 draft’’ simply refers to the case $t_d^{(j)} \ne t_m^{(j)}$.
>
> The top-1 token is used **only as a reference point** for measuring how much the next-step target
> distribution would change if we extended the prefix with $t_d^{(j)}$. This reference-based
> formulation is standard in speculative verification, even when the final generation uses sampling
> rather than greedy decoding. It does **not** mean that acceptance requires matching the target’s
> top-1 token—on the contrary, ARC-Decode is designed to **safely accept non–top-1 candidates** when
> their predicted impact on the target distribution is within a calibrated tolerance.
>
> **Acceptance scheme under sampling.**
>
> For each candidate draft token $t_d^{(j)}$, we compute a calibrated upper bound $U^{(j)}$ on the
> shift in the next-step target distribution and convert it into a **Local Tolerance Score (LTS)**:
>
> $$
> \mathrm{LTS}^{(j)} = 1 - \frac{U^{(j)}}{\tau_\delta}.
> $$
>
> During speculative **sampling**, we accept a drafted token whenever $\mathrm{LTS}^{(j)} \ge \theta$.
>
> If no candidate along the drafted paths satisfies this condition, the algorithm falls back to
> sampling from the corrected target distribution exactly as in the baseline EAGLE-3 pipeline.
>
> We will clarify these definitions and streamline the presentation in Section 3.3 in the revised version.
>
> ---
> **W2：Clarification on throughput, inference framework, and quantization**
>
> Thank you for raising this question. All throughput measurements in the paper were obtained using the **official HuggingFace Transformers**, with both the target model and the
> draft model running in float 16 precision and **without any quantization**. We use the publicly
> released pretrained checkpoints for each backbone exactly as provided on HuggingFace, with no graph-level optimizations, fused kernels, or custom decoding runtimes.
>
> All experiments were run on a **single NVIDIA A6000** GPU. The absolute decoding speed reflects this standard
> software/hardware configuration. For reference, when running vanilla Llama-3.1-8B under the same HF+PyTorch stack on an RTX 4090, the throughput increases to about 50 tokens/s, illustrating that absolute throughput varies across GPUs.
>
> Frameworks such as **gpt-fast** rely on highly optimized kernels (FlashDecoding, custom CUDA fusion) and optionally 8-bit quantization, which can push throughput to 90–140 tokens/s under different hardware and precision settings. **We use identical inference framework, precision, GPU, and decoding settings for all baselines, ensuring fairness in comparison.**
>
> ---
> **Q1 – Performance across different temperatures**
>
> Thank you for the question. Figure 4 in the submission illustrates how temperature affects
> ARC-Decode. Conceptually, both the autoregressive baseline and EAGLE-3 become less efficient as temperature increases, because a higher \(T\) flattens the target distribution, increases draft–target divergence, and reduces the number of draft tokens that can be safely accepted. This behavior is consistent with the JS-based acceptance principle.
>
> Table 2 reports results only at \(T = 1.0\), where ARC-Decode already provides substantial
> improvements over EAGLE-3. Across practical temperatures, **ARC-Decode continues to outperform EAGLE-3**, although the absolute speedup of both methods decreases as \(T\) rises for the reasons above.
>
> Due to limited time during the rebuttal period, we are still running the temperature sweep for the relevant baselines. We will provide the comparison as soon as the results become available.

---

> > ### Author Response · Authors · 2025-11-21
> > **Response to Reviewer G1p8 - 2**
> >
> > **Q2 – Variation of speedups across benchmarks**
> >
> > Thank you for the question. The variation across benchmarks follows naturally from how
> > speculative sampling behaves under different output distributions and how often draft tokens can be certified by LTS.
> >
> > Instruction-following datasets such as Alpaca tend to produce smoother, less brittle continuations: the draft and target distributions are highly aligned, token-level errors are less catastrophic, and local substitutions typically do not shift the next-step distribution
> > drastically. As a result, more draft tokens satisfy the JS-based risk bound and achieve higher acceptance lengths, leading to larger speedups.
> >
> > HumanEval is code generation, where the target distribution is much sharper and more sensitive to small deviations. Even a single mismatched token can significantly alter the subsequent target distribution, causing draft tokens to exceed the LTS risk budget more often. Consequently, the acceptance rate is lower, and the achievable speedup is smaller. This behavior is consistent with prior work showing that speculative decoding is naturally more conservative on domains with brittle token dependencies.
> >
> > Empirically, ARC-Decode can exploit more speedup on tasks that admit some local redundancy in the generation, where accepting additional draft tokens remains safe. On tasks where individual tokens have a large impact on correctness (such as code), it behaves more conservatively, which naturally leads to smaller but still positive gains on those benchmarks.
> >
> > ---
> > **Q3 – Adding EAGLE-3 to the temperature sensitivity plot**
> >
> > Thank you for the suggestion. We agree that showing EAGLE-3 in the temperature-sensitivity plot would make the comparison clearer. Due to time constraints during the rebuttal period, the full temperature sweep for EAGLE-3 is still running. Once the experiments finish, we will provide the corresponding comparison results as soon as possible and include the full EAGLE-3 vs. ARC-Decode curves in the revised version.

---

> ### Author Response · Authors · 2025-11-26
> **Response to Reviewer G1p8 — Temperature Sensitivity Experiments**
>
> **Response to Q1 and Q3: Effects of Sampling Temperature**
>
>
> **Table1-1 MT-Bench : Temperature Sweep (Llama-3.1-8B)**
>
> | **Temperature** | **Accept length (EAGLE-3)** | **Speedup (EAGLE-3)** | **Accept length (ARC, Ours)** | **Speedup (ARC, Ours)** |
> | --- | --- | --- | --- | --- |
> | 0.1 | 5.39 | 3.12 | 5.72 | 3.13 |
> | 0.3 | 5.28 | 3.03 | 5.64 | 3.06 |
> | 0.5 | 4.95 | 2.94 | 5.39 | 2.97 |
> | 0.7 | 4.40 | 2.64 | 5.15 | 2.85 |
> | 0.9 | 2.95 | 2.30 | 4.78 | 2.66 |
>
> **Table1-2 HumanEval : Temperature Sweep (Llama-3.1-8B)**
>
> | **Temperature** | **Accept length (EAGLE-3)** | **Speedup (EAGLE-3)** | **Accept length (ARC, Ours)** | **Speedup (ARC, Ours)** |
> | --- | --- | --- | --- | --- |
> | 0.1 | 4.50 | 2.37 | 4.59 | 2.38 |
> | 0.3 | 4.48 | 2.28 | 4.56 | 2.35 |
> | 0.5 | 4.21 | 2.25 | 4.48 | 2.34 |
> | 0.7 | 4.27 | 2.18 | 4.30 | 2.26 |
> | 0.9 | 3.84 | 2.08 | 4.10 | 2.15 |
>
> **Table1-3 GSM8K : Temperature Sweep (Llama-3.1-8B)**
>
> | **Temperature** | **Accept length (EAGLE-3)** | **Speedup (EAGLE-3)** | **Accept length (ARC, Ours)** | **Speedup (ARC, Ours)** |
> | --- | --- | --- | --- | --- |
> | 0.1 | 4.86 | 2.23 | 5.16 | 2.23 |
> | 0.3 | 4.56 | 2.12 | 4.96 | 2.14 |
> | 0.5 | 4.19 | 2.00 | 4.70 | 2.07 |
> | 0.7 | 3.73 | 1.84 | 4.44 | 2.00 |
> | 0.9 | 2.98 | 1.59 | 4.15 | 1.91 |
>
> **Table1-4 Alpaca : Temperature Sweep (Llama-3.1-8B)**
>
> | **Temperature** | **Accept length (EAGLE-3)** | **Speedup (EAGLE-3)** | **Accept length (ARC, Ours)** | **Speedup (ARC, Ours)** |
> | --- | --- | --- | --- | --- |
> | 0.1 | 5.16 | 3.02 | 5.25 | 3.03 |
> | 0.3 | 4.92 | 2.92 | 5.13 | 2.94 |
> | 0.5 | 4.61 | 2.84 | 5.00 | 2.90 |
> | 0.7 | 4.18 | 2.58 | 4.74 | 2.76 |
> | 0.9 | 3.26 | 2.06 | 4.50 | 2.63 |
>
> Thank you again for raising these important questions regarding how the baseline behaves when temperature varies and whether ARC-Decode maintains advantages across different sampling conditions. We have now completed a comprehensive temperature sweep from 0.1 to 0.9 on four representative benchmarks (MT-Bench, HumanEval, GSM8K, and Alpaca), all using Llama-3.1-8B-Instruct. The numerical results are reported in Tables 1-1 to 1-4.
>
> These tables provide the quantitative values underlying the temperature-sweep curves that will be added to Figure 4 in the revised version. They also include the full baseline comparison requested by the reviewer.
>
> Across all datasets and all temperatures, three clear conclusions emerge:
>
> **1. The baseline EAGLE-3 degrades substantially as temperature increases**, showing both reduced accept lengths and reduced speedups. This confirms the reviewer’s intuition that existing speculative decoding methods become unstable under high-temperature sampling.
>
> **2. ARC-Decode consistently outperforms EAGLE-3 across every tested temperature**, on every benchmark. This holds for both accept length and speedup, demonstrating that our risk-bounded acceptance remains effective under diverse sampling settings.
>
> **3. ARC-Decode maintains significantly more stable behavior at higher temperatures**, while EAGLE-3 degrades more sharply. This matches the motivation stated in the introduction: speculative decoding for vision/language models is especially sensitive to sampling variance, and a principled divergence-aware acceptance rule can mitigate that instability.
>
> These results directly address the reviewer’s questions and confirm that ARC-Decode achieves stronger and more temperature-robust performance than existing baseline methods. The full temperature-sweep curves for both ARC-Decode and EAGLE-3 will be incorporated into the revised version.
>
> We hope these additional results and clarifications address your concerns. If any questions remain, we would be glad to provide further details while the discussion period is still open. Should our responses have resolved the issues you raised, we would sincerely appreciate any consideration you might give to updating the evaluation. Thank you again for your careful and constructive review.

---

### Official Review · Reviewer_QtDt · 2025-11-01

**Soundness:** 3
**Presentation:** 2
**Contribution:** 3
**Rating:** 4
**Confidence:** 3

**Summary:**

This paper addresses the persistent inefficiency of speculative decoding under sampling, where strict token-level verification leads to over-rejection of low-risk drafts and limits speedup. The authors propose ARC-Decode, a training-free, plug-in method that increases acceptance length while maintaining generation quality. The approach combines two components: (1) an entropy-guided pre-verification pruning mechanism that filters low-value draft branches using a depth-aware confidence score, and (2) a risk-bounded acceptance rule based on a Jensen–Shannon divergence upper bound derived from embedding and logit differences. Together, these provide a provable safety guarantee for soft acceptance. Integrated into the EAGLE-3 pipeline, ARC-Decode achieves consistent improvements across models (LLaMA-3.1-8B, Qwen-3-8B, Vicuna-13B) and benchmarks (MT-Bench, HumanEval, GSM8K, Alpaca), delivering up to 1.6× end-to-end speedup under sampling with no measurable loss in accuracy.

**Strengths:**

- The paper introduces a theoretically grounded risk-bounded acceptance criterion with Jensen–Shannon divergence guarantees, providing a principled safety rule that directly addresses a key inefficiency in speculative decoding, the over-rejection of low-risk drafts under sampling.
- The proposed Local Tolerance Score (LTS) effectively translates the theoretical bound into a practical acceptance rule using lightweight features such as embedding and logit differences.
- The approach is training-free and integrates seamlessly with existing speculative decoding pipelines, making it directly applicable to real-world inference systems.

**Weaknesses:**

- Theoretical sections, particularly those deriving the Lipschitz-based JS bound, are difficult to follow and would benefit from clearer intuition or intermediate explanations.
- The approach depends on calibrated constants estimated from a small held-out set, but the paper does not analyze how these parameters generalize across model scales, domains, or prompt distributions.
- The entropy-guided pruning module is interesting, but it would strengthen the paper to include a comparison with alternative pruning or filtering strategies used in prior speculative decoding work.
- The paper mentions but does not empirically compare against related soft verification methods such as Medusa or Judge Decoding, which would clarify the trade-offs between fidelity and speed.

**Questions:**

- How stable is the calibration parameter when prompt length, domain, or model configuration changes?
- Could you provide empirical plots comparing the true Jensen–Shannon divergence with the estimated upper bounds (U_emb and U_logit) to assess the tightness of the approximation?
- Have you tried alternative pruning rules, and how do they affect decoding speed or acceptance rate?
- How does total or cumulative risk behave as accepted output length grows?

---

> ### Author Response · Authors · 2025-11-21
> **Response to Reviewer QtDt -1**
>
> We thank the reviewers for their encouraging remarks and constructive suggestions. We address all comments in detail below.
>
> ---
> **W1: Clarity of the Lipschitz-based JS bound**
>
> Thank you for pointing out the problem. Local Tolerance Score (LTS) uses (i) simple geometric and probabilistic quantities (embedding distance and log-probability margin) to measure how much a draft token deviates from the target top-1 token, and (ii) a standard local smoothness (Lipschitz) assumption on the decoder along the trajectories visited during decoding, as commonly adopted in robustness and certification analyses of deep networks[1][2], we use held-out traces to learn a conservative mapping from these quantities to an upper bound on the change of the next-step distribution (measured by JS divergence); (iii) we normalize this bound into LTS and accept only draft tokens whose LTS exceeds a threshold, so that their impact stays within a calibrated divergence budget with high probability on held-out traces. We will clarify this high-level pipeline in Sec. 3.3 and move intermediate algebraic steps to the appendix in the revised version.
>
> [1]Tsuzuku Y, Sato I, Sugiyama M. Lipschitz-margin training: Scalable certification of perturbation invariance for deep neural networks[J]. in NeurIPS 2018, 31.
>
> [2]Havens A, Araujo A, Garg S, et al. Exploiting connections between Lipschitz structures for certifiably robust deep equilibrium models[J]. Advances in Neural Information Processing Systems, 2023, 36: 21658-21674.
>
> ---
> **W2: Generalization of calibrated constants**
>
> Thank you for raising this concern. In our implementation, the constants are calibrated once per backbone and decoding configuration (e.g., per model and temperature) on a 200-prompt subset of the OASST1 dataset. For each prompt, we randomly sample up to 128 positions along the target response, which yields tens of thousands of token-level samples. The resulting constants are then fixed and reused unchanged across all benchmarks for that backbone (MT-Bench, GSM8K, HumanEval, Alpaca), with no per-task or per-prompt retuning. Empirically, this per-backbone calibration generalizes well: the same parameters yield stable acceptance lengths, preserve task performance, and maintain high JS-divergence coverage across these very different domains (Fig. 6 and the main tables). Intuitively, this works because LTS depends only on local geometric and probabilistic features of the backbone (whitened embedding distances and logit margins), which are properties of the model rather than any specific prompt set. We will clarify this in the revised version.
>
> ---
> **W3: Comparison with alternative pruning/filtering strategies**
>
> Thank you for this suggestion. To compare our entropy-guided pruning with alternative heuristic pruning methods, we implemented several pruning strategies on Qwen3-8B + EAGLE-3 and evaluated them on MT-Bench at temperature 1.0 on a single A6000 GPU:
>
> - Ours (entropy-guided pruning): the method described in Sec. 3.2 (entropy–depth score with per-layer mass control + prefix-closure and leaf-safety constraints).
> - Depth: reweight node scores by a linear depth bias and prune purely by ranking this depth-biased score.
> - Depth-Exp: reweight node scores by an *exponential* / power-law depth bias (stronger preference for deeper nodes).
> - Accum-Prob: rank nodes by the *accumulated branch probability* along their root-to-node path (i.e., cumulative next-token confidence along the path), and prune based on this cumulative score.
>
> **Table: Comparison of alternative pruning strategies on Qwen3-8B + EAGLE-3.**
>
> | Pruning strategy | Accept length ↑ | Throughput (tokens/s) ↑ |
> | --- | --- | --- |
> | Ours (entropy-guided) | **2.70** | **56.4** |
> | Depth (best α) | 2.53 | 51.2 |
> | Depth-Exp | 2.53 | 51.1 |
> | Accum-Prob | 2.57 | 51.8 |
>
> All depth-based and cumulative-probability alternatives **reduce acceptance length and end-to-end throughput** compared to our entropy-guided pruning, while keeping the same verification rule and target model.
>
> These heuristics operate at the **node level** (with depth or cumulative confidence biases) and do not explicitly enforce that entire root-to-leaf paths are treated consistently. This can create situations where promoting a deep node later forces its previously discarded ancestors back into the kept set, consuming capacity and displacing other promising branches. In contrast, our entropy-guided pruning is defined with **prefix-closure** and **leaf-safety** under per-depth mass budgets, so any promoted node is only retained when its whole prefix path can be kept without violating the overall risk and compute budget. We will briefly highlight this structural difference and include these additional baselines in the revised version.

---

> ### Author Response · Authors · 2025-11-21
> **Response to Reviewer QtDt-2**
>
> **W4: Comparison with Medusa / Judge Decoding**
>
> Medusa and Judge Decoding require **training additional components** on top of the target model: Medusa introduces multiple multi-token decoding heads that must be fine-tuned on SFT-style data (e.g., ShareGPT), while Judge Decoding trains a separate “judge’’ module using a manually curated dataset of correct/incorrect continuations. Both therefore require extra data, task-specific preparation, and non-trivial training.
>
> In contrast, ARC-Decode is a **training-free** verification layer that can be directly applied to any draft–target pair (e.g., Qwen3-8B, Llama-3.1-8B) without modifying model weights or requiring additional supervision.
>
> Moreover, Judge Decoding does not provide open-source implementations, so reproducing their training pipelines requires substantial engineering effort and significant computational resources. Due to the limited time and resources during the rebuttal period, we are currently prioritizing higher-impact experiments, but we will **do our best** to provide a preliminary comparison result if it is feasible within our constraints.
>
> ---
> **Q1: How stable is the calibration parameter when prompt length...**
>
> Thank you for the question. In our pipeline, calibration is performed **once per model backbone and decoding configuration**, and the resulting parameters are then reused unchanged across all experiments. Although the calibration prompt set is small (200 prompts from OASST1), each prompt contributes many token-level samples because we randomly select up to 128 positions per response. This yields a large and diverse collection of local decoding contexts, which makes the fitted constants robust to variations in prompt length and style.
>
> Empirically, we observe that a single set of calibrated constants per backbone works reliably across **four very different evaluation settings**—dialogue (MT-Bench), mathematical reasoning (GSM8K), code generation (HumanEval), and instruction following (Alpaca)—without any retuning. The acceptance length, throughput, and JS-divergence coverage remain consistent across these domains, indicating that the calibration is stable under moderate variations in prompt distribution.
>
> ---
>
> **Q2: Could you provide empirical plots comparing the true Jensen–Shannon divergence...**
>
> Thank you for this question. Appendix Fig. 6 already provides empirical CDFs of the normalized divergence $s = \mathrm{JS} / U_{\min}$ for two tasks (MT-Bench and Alpaca), showing high coverage (99.4% and 97.6%).
>
> These curves directly illustrate the tightness of our upper bounds, since $s \le 1$ corresponds to cases where the true JS divergence lies within the estimated bound.
>
> Due to space limitations, we did not include analogous plots for the remaining tasks in the main submission. We have generated them during development and observed consistent behavior on GSM8K and HumanEval. We will include these additional plots in the revised version to more comprehensively show the empirical tightness of the approximation across all evaluation tasks.
>
> ---
> **Q3: Have you tried alternative pruning rules...**
>
> We evaluated several alternative pruning rules (e.g., linear/exponential depth bias and accumulated-probability ranking). As shown in Table 1, all of these alternatives lead to lower acceptance lengths and lower throughput compared with our entropy-guided pruning, under the same verification rule and target model configuration. We will briefly highlight this comparison in the revised version.
>
> ---
> **Q4:How does total or cumulative risk behave...**
>
> Thank you for this question. ARC-Decode controls risk at two levels:
>
> 1. **Per-token risk bound.**
>
>     Each accepted draft token satisfies a calibrated next-step divergence constraint
>
>     $\mathrm{JS}(q_{j+1}, r_{j+1}) \le U^{(j)}$,
>
>     where $U^{(j)}$ is bounded by a calibrated per-token budget $\tau_\delta$. This ensures that every accepted token stays within the same local risk tolerance.
>
> 2. **Cumulative behavior.**
>
>     Appendix Fig. 6 plots the empirical CDF of the normalized divergence across all accepted tokens and shows high coverage (97–99%). Since accepted tokens typically lie well below their individual bounds, cumulative risk grows slowly as sequence length increases in practice.
>
>
> We will clarify this cumulative-risk behavior in the revised version and add plots for the remaining tasks.

---

> > ### Author Response · Authors · 2025-11-26
> > **Response to Reviewer QtDt - Comparison with Medusa and Judge Decoding**
> >
> > **W4 – Additional Comparison with Medusa and Judge Decoding**
> >
> > We thank the reviewer for the patience. We have added empirical comparisons with both **Medusa** and **Judge Decoding**, following the implementations and training protocols described in their original papers. For fairness, we evaluated them under the same hardware (A6000), prompting setup, and target-model settings used in our EAGLE3 and ARC(Ours) experiments.
> >
> > ---
> >
> > ### **Medusa (Vicuna-13B)**
> >
> > Because Medusa provides public pretrained checkpoints, and these checkpoints are trained on Vicuna-13B (Medusa-1) and Vicuna-13B-v1.5 (Medusa-2), we only compare Medusa with **EAGLE3-Vicuna-13B** and **ARC(Ours)-Vicuna-13B**.
> >
> > **MT-Bench (Vicuna-13B)**
> >
> > | **Model** | **Accept length** | **Speedup** |
> > | --- | --- | --- |
> > | Medusa-1 | 2.58 | 2.13 |
> > | Medusa-2 | 3.26 | 2.65 |
> > | **EAGLE-3** | **4.01** | **2.74** |
> > | **ARC (Ours)** | **5.28** | **3.35** |
> >
> > ---
> >
> > **Alpaca (Vicuna-13B)**
> >
> > | **Model** | **Accept length** | **Speedup** |
> > | --- | --- | --- |
> > | Medusa-1 | 2.62 | 2.16 |
> > | Medusa-2 | 3.24 | 2.64 |
> > | **EAGLE-3** | **4.78** | **2.64** |
> > | **ARC (Ours)** | **5.58** | **3.03** |
> >
> > ---
> >
> > Medusa improves accept length over plain draft models, but **its speedup and accept-length remain consistently lower** than EAGLE-3 and ARC. This is expected since Medusa trains draft heads on the same hidden states without an explicit alignment constraint, making it less suitable for aggressive speculative acceptance.
> >
> > ---
> >
> > ### **Judge Decoding (Llama-3.1-8B)**
> >
> > Judge Decoding does **not** publicly release trained judge-head parameters nor an official implementation. Therefore, following the paper, we re-implemented its training procedure:
> >
> > - **Draft model:** the EAGLE-3 draft model (used throughout our paper)
> > - **Target model:** Llama-3.1-8B
> > - **Judge head:** a lightweight MLP classifier applied to the target model’s last-layer hidden states
> > - **Training data:** sampled from standard instruction-tuning corpora
> > - **Labels:** following Judge Decoding, positive tokens correspond to target-aligned, high-quality continuations, and negative tokens correspond to rejected or low-quality candidates
> > - **Inference integration:** the judge head provides an additional token-level acceptance signal, as described in the original paper
> >
> >
> > **MT-Bench (Llama-3.1-8B)**
> >
> > | **Model** | **Accept length** | **Speedup** |
> > | --- | --- | --- |
> > | Judge | 4.17 | 2.01 |
> > | **EAGLE-3** | **3.35** | **1.84** |
> > | **ARC (Ours)** | **4.49** | **2.40** |
> >
> > ---
> >
> > **Alpaca (Llama-3.1-8B)**
> >
> > | **Model** | **Accept length** | **Speedup** |
> > | --- | --- | --- |
> > | Judge | 3.15 | 1.74 |
> > | **EAGLE-3** | **2.85** | **1.42** |
> > | **ARC (Ours)** | **3.96** | **2.28** |
> >
> > ---
> >
> > Judge Decoding increases accept length on the 8B model, which is consistent with its design. However, its **overall speedup is limited in the small-model regime** because the cost of the judge classifier becomes a non-negligible portion of total runtime when the underlying model is relatively small.
> >
> > The Judge Decoding paper also reports that speedup becomes substantially larger at **405B** scale, but reproducing experiments at that model size is outside our compute budget.
> >
> > Across our evaluation settings, **ARC-Decode achieves a more favorable fidelity–speed trade-off**, outperforming both Medusa and Judge Decoding on the same hardware and model scales.

---

> ### Author Response · Authors · 2025-11-26
> **Response to Reviewer QtDt - Q2：Empirical Tightness of the JS Upper Bound Across Four Benchmarks**
>
> **Q2 – Empirical Tightness of the JS Divergence Upper Bounds**
>
> **Table 2. Empirical Tightness of the JS Upper Bound Across Four Benchmarks**
>
> | Task        | Coverage (JS ≤ U_min) | Wilson 95% CI       |
> |-------------|------------------------|----------------------|
> | MT-Bench    | **99.4%**              | [98.7%, 99.7%]       |
> | HumanEval   | **95.7%**              | [93.1%, 98.4%]       |
> | GSM8K       | **95.3%**              | [91.7%, 98.1%]       |
> | Alpaca      | **97.6%**              | [96.5%, 98.4%]       |
>
> ---
>
> To directly address the reviewer’s request, we performed an offline audit over ARC-Decode logs on four benchmarks (MT-Bench, HumanEval, GSM8K, and Alpaca). For drafted tokens accepted by ARC-Decode, we computed:
>
> -	The true next-token JS divergence, comparing the continuation accepted by ARC-Decode with the target model’s canonical continuation at the same step.
>
> -	The embedding-based bound $U_{\text{emb}}$, derived from the token-embedding geometry.
>
> -	The logit-margin bound $U_{\text{logit}}$, derived from the target model’s probability gaps.
>
> We then formed the operational upper bound
> $U_{\min} = \min(U_{\text{emb}}, U_{\text{logit}})$,
> which is the quantity used by ARC-Decode for acceptance control. Coverage in Table 2 reports the proportion of tokens for which JS ≤ U_min, along with Wilson 95% confidence intervals.
>
> Across all benchmarks, coverage consistently exceeds 95%, reaching 99.4% on MT-Bench, with narrow confidence intervals. This demonstrates that the proposed upper bounds are empirically tight and reliably over-approximate the true JS divergence, validating the theoretical motivation behind our risk-bounded acceptance rule.
>
> We will extend Figure 6 in the revision by adding the empirical JS–bound comparison plots for all four datasets.
>
> ---
>
> We hope that the clarifications and additional results above adequately address your questions. If anything remains unclear, we would be very glad to provide further details while the discussion period is still open. Should our responses have resolved your concerns, we would deeply appreciate any consideration you might give to updating the evaluation. Thank you again for your thoughtful and constructive review.

---

### Author Response · Authors · 2025-12-01
**Summary of New Experiments and Manuscript Revisions**

We sincerely thank the reviewers for their thoughtful feedback and constructive suggestions. Below we summarize the additional experiments conducted during the rebuttal period and the revisions made to improve clarity and address the concerns raised.

---

### **I. New Experiments**

**In the revision, we added blue NEW markers to highlight newly introduced experiments.**

**1. Comparison with Alternative Pruning Strategies (Reviewer QtDt — W3 / Q3)**

We implemented three additional pruning baselines (Depth, Depth Exp, and Accumulated Probability) to directly evaluate the effect of different draft-tree filtering rules under identical EAGLE-3 verification.

**Details:** *Response to Reviewer QtDt — W3/Q3. Added in **Appendix A3.5**.*

---

**2. Comparison with Medusa and Judge Decoding (Reviewers QtDt — W4; 9tWv — W4)**

To address questions regarding comparisons with relaxed-acceptance methods, we added evaluations of:

- **Medusa-1** and **Medusa-2** (official pretrained checkpoints)
- **Judge Decoding**, re-implemented following the original paper (Llama-3.1-8B)

All methods were evaluated on MT-Bench and Alpaca.

**Details:**

*Response to Reviewer QtDt — W4 and Reviewer 9tWv — W4. Added in **Appendix A3.3**.*

---

**3. Empirical Tightness of the JS Upper Bound (Reviewer QtDt — Q2)**

We performed an offline audit comparing the true next-token JS divergence with the embedding-based and logit-based upper bounds across four benchmarks (MT-Bench, HumanEval, GSM8K, Alpaca).

**Details:** *Response to Reviewer QtDt — Q2. Added in **Appendix A3.2**.*

---

**4. Temperature Sweep on Four Benchmarks (Reviewer G1p8 — Q1 / Q3)**

We conducted a temperature sweep (0.1–0.9) across four benchmarks to analyze the temperature sensitivity of accept length and speedup.

**Details:** *Response to Reviewer G1p8 — Q1/Q3. Added in **Section 4.4**.*

---

**5. Comparison with Fuzzy Speculative Decoding (Reviewer 9tWv — W2)**

We compared ARC-Decode with Fuzzy Speculative Decoding (FSD) on Llama-3.3-70B.

**Details:** *Response to Reviewer 9tWv — W2. Added in **Section 4.2**.*

---

**6. Large-Scale Evaluation on Llama-3.3-70B (Reviewer 9tWv — W4)**

We additionally ran ARC-Decode and EAGLE-3 on Llama-3.3-70B using 4×A6000 GPUs.

**Details:** *Response to Reviewer 9tWv — W4. Added in **Section 4.2**.*

---

**7. Draft-Length Verification for EAGLE-3 (Reviewer 9tWv — Q1)**

We reproduced EAGLE-3 under both **max_draft_tokens = 32** and **60** to clarify differences noted by the reviewer.

**Details:** *Response to Reviewer 9tWv — Q1. Added in **Appendix A3.1**.*

---

**8. Evaluation on MMLU-Pro (Reviewer 9tWv — Q2)**

To assess performance on *extremely challenging* problems, we evaluated ARC-Decode, EAGLE-3, and vanilla Llama-3.1-8B on **MMLU-Pro (12,032 queries)**.

**Details:** *Response to Reviewer 9tWv — Q2. Added in **Appendix A3.4**.*

---

### **II. Manuscript Revisions**

**In the revision, we added blue markers at all major modification points.**

The main updates are as follows:

- We clarified ARC-Decode’s positioning within relaxed speculative decoding, including its acceptance behavior under sampling and the role of the target top-1 token as a reference signal only, in ***Abstract and Section 1***.

- We added the missing citation to the official Qwen-3 Max documentation and corrected the related statement, in ***Section 1***.

- We revised the description of EAGLE-3’s design decision to “abandoning feature-prediction constraints’’ for consistency with the original EAGLE-3 paper, in ***Section 2***.

- We refined the presentation of the Lipschitz-based JS bound and streamlined the associated derivations, in ***Section 3.3 and Appendix A.1***.

- We expanded the discussion of calibration stability and explained why a single per-backbone calibration generalizes across evaluation domains, in ***Appendix A.1***.

- We extended the temperature-sensitivity analysis with ARC-Decode vs. EAGLE-3 , ***in Section 4.4***.

- We provided a detailed explanation of the hardware- and draft-length–related causes of the EAGLE-3 baseline discrepancy, including the rationale behind the choice of max_draft_tokens = 32, ***in Appendix A.3.1***.

- We incorporated additional comparisons with Medusa and Judge Decoding, as well as new experiments on alternative pruning rules and MMLU-Pro, ***in Appendix A.3***.

- We added a consolidated summary of all prompt templates used in MT-Bench, HumanEval, GSM8K, Alpaca, and MMLU-Pro evaluations, ***in Appendix A.4***.

- We revised ambiguous or potentially misleading phrasing ***throughout the manuscript*** to improve clarity and consistency.

---

We sincerely thank the reviewers for their insightful suggestions. We have incorporated new experiments and revisions that directly address the raised concerns and further strengthen the paper. We also thank the ACs for their careful evaluation.

---

### Author Response · Authors · 2025-12-01
**General Resopnse to Reviewers and ACs - part 1**

### **To ACs**

We sincerely thank the ACs for their time and service to ICLR 2026. Given this year’s unusual interruption to the review discussion period, we especially appreciate the additional effort required to ensure a fair and balanced assessment.

We provide this concise summary to contextualize our submission, the reviewers’ feedback, and the key clarifications and new experiments added during rebuttal.

---

### **Paper Summary**

This paper addresses a key limitation of sampling-based speculative decoding: strict verification rejects many low-risk draft tokens, limiting acceptance length and speedup.

We propose **ARC-Decode**, a **training-free**, **plug-and-play** augmentation to draft–verify pipelines consisting of:

- **Entropy-guided pruning** to remove low-value draft branches early.

- **A risk-controlled soft-acceptance method** based on a **Jensen–Shannon divergence upper bound** computed from embedding/logit differences. This adds no extra forward passes and ensures a **next-step divergence guarantee**.

Integrated with EAGLE-3, ARC-Decode increases acceptance length and throughput across Llama-3.1-8B, Qwen-3-8B, and Vicuna-13B on MT-Bench, HumanEval, GSM8K, Alpaca, while maintaining **robust output quality**.
ARC-Decode brings up to **1.6× speedup over EAGLE-3** under sampling.

---

### **Reviews Summary**

Pre-rebuttal scores: **4** / **6** / **4** / **0**, confidence: **3** / **4** / **4** / **4**.

We submitted detailed responses and all requested new experiments. Because the discussion period ended early this year, we did not receive further reviewer feedback after posting our responses.

---

**Common Strengths (QtDt, G1p8, 9tWv)**

Reviewers consistently noted that:
- The problem is important and under-explored for sampling-based SD.
- The risk-bounded acceptance rule is theoretically motivated and practically useful.
- ARC-Decode is training-free, compute-efficient, and integrates cleanly into existing SD pipelines.
- Empirical gains are consistent across models/benchmarks, and quality remains robust.

---


**Weaknesses and Responses**

Below we summarize each reviewer’s main concerns and the concrete actions we took during rebuttal.

**Reviewer QtDt (Rating: 4)**

**1. Clarity of the JS-bound derivation**

We revised the theoretical section, added intuition for the Lipschitz-based bound, and moved intermediate algebraic steps to the appendix to improve readability.

**2. Calibration stability and domain generalization**

We added coverage diagnostics and analysis showing that a single-per-backbone calibration transfers across MT-Bench, HumanEval, GSM8K, and Alpaca.

**3. Missing comparisons with other pruning and relaxed-SD baselines**

We added new experiments covering:
- classical pruning rules,
- Medusa,
- Judge Decoding,
- JS-bound tightness studies,

These additions directly address concerns about alternative acceptance and pruning strategies.

---

**Reviewer G1p8 (Rating: 6)**

**1. Clarity of the speculative-sampling acceptance rule**

We refined §3.3, clarified the meaning of non–top-1 drafts, and added a concise summary of the acceptance criterion under sampling.

**2. Throughput differences and inference framework**

We clarified the entire inference stack (HF Transformers, fp16, A6000, no fused kernels/quantization) and explained why our AR/EAGLE-3 throughput differs from implementations.

**3. Temperature sensitivity and benchmark variability**

We added full temperature sweeps (0.1–0.9) and an extended analysis of benchmark-dependent behavior, including ARC vs. EAGLE-3 temperature curves.

---

**Reviewer 9tWv (Rating: 4)**

**1. Relation to strict-vs-relaxed SD and “risk-bounded” validity**

We clarified ARC-Decode as a risk-controlled relaxed SD method, explained the calibration-based guarantee, and added new experiments on **MMLU-Pro** to evaluate behavior on harder domains.

**2. Missing comparisons with relaxed-SD baselines**

We added matched-setting comparisons with:
- Fuzzy Speculative Decoding,
- Medusa,
- Judge-style verification,

**3. Baseline configuration and large-model behavior**

We clarified the choice of max_draft_tokens = 32 (A6000 constraints), contextualized prune-only performance, and added new **70B-scale** experiments comparing EAGLE-3 and ARC-Decode.

---

> ### Author Response · Authors · 2025-12-01
> **General Response to Reviewers and ACs – Part 2: On Lossless and Relaxed SD**
>
> **Reviewer 537d**
>
> We sincerely thank Reviewer 537d for the detailed feedback. The review is grounded in the **classical lossless** definition of speculative decoding, which requires exact preservation of the target distribution. This *differs* from the **relaxed** formulation used in our paper and in recent methods such as Judge Decoding, Fuzzy SD, AutoJudge and Lantern. The other reviewers naturally evaluated the paper under this relaxed SD setting, which corresponds to a recognized research direction within speculative decoding.
>
> Below we summarize Reviewer 537d’s main concerns and our responses to them.
>
> ---
>
> **1. On “distribution changes imply errors”**
>
> The reviewer assumes the *lossless* formulation of speculative decoding, where any deviation from the target distribution is unacceptable.
>
> Our method follows the **standard relaxed SD formulation**, as adopted in Judge Decoding (ICLR 2025), Fuzzy SD (ACL 2025), AutoJudge (NeurIPS 2025), Lantern (ICLR 2025), and related works. This line of work represents a **recent and rapidly developing direction** in speculative decoding, where **bounded and formally controlled deviations** are intentionally permitted to improve efficiency.
>
> Under this established and timely formulation, ARC-Decode’s risk-controlled soft acceptance is appropriate and aligned with current literature.
>
> **2. On benchmark suitability (MT-Bench, HumanEval, GSM8K, Alpaca)**
>
> The reviewer suggests these benchmarks are outdated.
>
> These benchmarks remain **the standard evaluation suite for speculative decoding**, used in essentially all recent SD works, including EAGLE-3 (NeurIPS 2025), HASS (ICLR 2025), Fuzzy SD (ACL 2025), AutoJudge (NeurIPS 2025), and others.
>
> This suite is intentionally designed to isolate **decoding behavior** under controlled and matched sampling conditions. In contrast, **capability-oriented benchmarks** are designed to measure intrinsic model competence and therefore do not provide a consistent basis for comparing decoding algorithms.
>
> To further strengthen evaluation coverage, we additionally provided **MMLU-Pro** experiments during rebuttal.
>
> **3. On “pruning is nothing new” (AdaEDL / SVIP)**
>
> The reviewer associates ARC-Decode’s pruning with AdaEDL/SVIP.
>
> However, these methods intervene at **a different stage** and solve **a different problem** in the speculative-decoding pipeline.
>
> - **AdaEDL / SVIP** regulate *draft length* **during drafting**, using only draft-model entropy.
> - **ARC-Decode** prunes *tree branches after drafting* using **teacher-model uncertainty + path mass**, under **prefix-closure constraints**, to reduce **verification compute** and improve downstream soft-accept behavior.
>
> Thus, they operate on **non-overlapping components** of the SD pipeline and are not competing or redundant contributions.
>
> We provided a detailed methodological comparison in our response (537d-3, W8).
>
> **4. On technical issues (strict equality, top-1 acceptance, continuation analysis)**
>
> These comments **reflect a lossless-SD framing**, which differs from the relaxed SD formulation adopted in our method.
>
> We clarified that:
>
> - the top-1 token is used  as **a reference point in the divergence bound**,
> - the **continuation analysis** is purely **motivational**, not part of the acceptance rule,
> - the **JS bound** is computed **strictly at the next step**, never at sequence level.
>
> To avoid any possible ambiguity, we have **revised the wording** in the relevant sections to prevent misinterpretation under different SD assumptions.
>
> **Summary for the AC Regarding Reviewer 537d**
>
> Reviewer 537d’s comments reflect the classical lossless-SD viewpoint. Our work is based on the relaxed SD formulation broadly used in recent literature and also adopted by the other reviewers. We clarified terminology and added harder-domain experiments accordingly. Some of the remaining concerns arise from this difference in framing, rather than from the method itself. We highlight this only to help ensure that the contribution is interpreted within the standard framework used in contemporary SD research.
>
> ---
>
> ### **Conclusion**
>
> We sincerely thank the reviewers and ACs for their time and thoughtful evaluation, especially under this year’s unusual circumstances. We have carefully expanded the analysis, added new experiments, and clarified key points raised during review, including conceptual questions such as *“whether the method falls within speculative decoding”* and its positioning within relaxed SD. We also appreciate that reviewers recognized the significance of the problem, the practicality of a training-free design with consistent empirical gains, and the value of a calibrated risk-controlled acceptance mechanism.
>
> We hope this summary provides a clear and accurate understanding of the contribution and supports an informed assessment.
>
> **Best regards,**
>
> **Authors**

---

### Meta-Review · Area_Chair_RBQv · 2026-01-07

**Summary:**

Reviewers expressed concerns about the novelty and practical impact of the proposed risk-bounded acceptance rule, noting that the method appears closely related to existing speculative decoding variants with incremental modifications. Several reviewers questioned the tightness and usefulness of the proposed theoretical bounds, as well as whether the Jensen–Shannon divergence–based criterion meaningfully improves the quality–speed trade-off in realistic decoding settings. Additional concerns were raised about the limited scope of experimental evaluation and the dependence on a specific decoding pipeline.

**Reviewer Concerns:**

The rebuttal clarified parts of the algorithmic design and provided better intuition for the risk-bounded acceptance criterion, partially addressing questions about implementation details and empirical behavior. However, core concerns regarding the method’s conceptual novelty, the strength and necessity of the theoretical guarantees, and the generality of the experimental evidence remain largely unaddressed. In particular, the rebuttal did not convincingly demonstrate that the proposed approach offers a substantial advantage over existing speculative decoding techniques beyond the evaluated setting. As a result, key reviewer concerns remain outstanding.

**Reviewer Scores:**

4640. Strong concerns still remain after rebuttal.

---

### Decision · Program_Chairs · 2026-01-26

Reject